# Greenland climate simulations show high Eemian surface melt which could explain reduced total air content in ice cores

Andreas Plach[1, 2, 3], Bo M. Vinther[4], Kerim H. Nisancioglu[1, 5], Sindhu Vudayagiri[4], and Thomas Blunier[4]

[1]Department of Earth Science, University of Bergen and Bjerknes Centre for Climate Research, Bergen, Norway
[2]Department of Meteorology and Geophysics, University of Vienna, Austria
[3]Climate and Environmental Physics, Physics Institute, University of Bern, Switzerland
[4]Centre for Ice and Climate, Niels Bohr Institute, University of Copenhagen, Denmark
[5]Centre for Earth Evolution and Dynamics, University of Oslo, Oslo, Norway

**Correspondence:** Andreas Plach (andreas.plach@gmail.com)

**Abstract.** This study presents simulations of Greenland surface melt for the Eemian interglacial period (~130000 to 115000 years ago) derived from regional climate simulations with a coupled surface energy balance model. Surface melt is of high relevance due to its potential effect on ice core observations, e.g., lowering the preserved total air content (TAC) used to infer past surface elevation. An investigation of surface melt is particularly interesting for warm periods with high surface melt, such as the Eemian interglacial period. Furthermore, Eemian ice is the deepest and most compressed ice preserved on Greenland, resulting in our inability to identify melt layers visually. Therefore, simulating Eemian melt rates and associated melt layers is beneficial to improve the reconstruction of past surface elevation. Estimated TAC, based on simulated melt during the Eemian, could explain the lower TAC observations. The simulations show Eemian surface melt at all deep Greenland ice core locations and an average of up to ~30 melt days year$^{-1}$ at Dye-3, corresponding to more than 600 mm water equivalent (w.e.) of annual melt. For higher ice sheet locations between 60 to 150 mm w.e. year$^{-1}$ on average are simulated. At the summit of Greenland this yields a refreezing ratio of more than 25 % of the annual accumulation. As a consequence, high melt rates during warm periods should be considered when interpreting Greenland TAC fluctuations as surface elevation changes. Additionally to estimating the influence of melt on past TAC in ice cores, the simulated surface melt could potentially be used to identify coring locations where Greenland ice is best preserved.

## 1 Introduction

The Eemian interglacial period (~130000 to 115000 years ago; thereafter ~130 to 115 ka) was the last period with a warmer-than-present summer climate on Greenland (CAPE Last Interglacial Project Members, 2006; Otto-Bliesner et al., 2013; Capron et al., 2014). Favourable orbital parameters (higher obliquity and eccentricity compared to today) during the early Eemian period caused a positive Northern summer insolation anomaly (and negative winter anomaly) at high latitudes, which led to a stronger seasonality (Yin and Berger, 2010). This stronger seasonality with relatively warm summer seasons is favourable for high melt rates across the Greenland ice sheet.

Unfortunately, the presence of surface melt can influence our ability to interpret ice core records. Measurements of $CH_4$, $N_2O$, and total air content (TAC) can be affected if melt layers are present. Other ice core measurements such as $\delta^{18}O$, $\delta D$, and deuterium excess appear to be only marginally affected (NEEM community members, 2013). However, refrozen melt has the potential to form impermeable ice layers (melt layers henceforth) that alter the diffusion of ice core signals.

The observed TAC of ice core records is the only direct proxy for past surface elevation of the interior of an ice sheet, i.e., the TAC is governed by the density of air which mainly decreases with elevation. However, TAC is also affected by low-frequency insolation variations (changing orbital parameters) at both Antarctic and Greenlandic sites (Raynaud et al., 2007; Eicher et al., 2016). Furthermore, Eicher et al. (2016) find a TAC response on millennial time scales (during Dansgaard-Oeschger-Events) which is hypothesed to be related to rapid changes in accumulation. While TAC can be estimated for each individual ice core without the need for other reference ice cores, another indirect method which has been applied to infer Holocene thinning of the Greenland ice sheet (Vinther et al., 2009) requires several ice cores. Vinther et al. (2009) compare the changes of $\delta^{18}O$ at coastal ice caps (stable surface elevation due to confined topography) with Greenland deep ice cores, and infer elevation changes. Unfortunately, Eemian ice core records are sparse, and therefore TAC is the only direct method available to estimate surface elevation changes this far back in time. Since the assumed surface elevation also influences the actual Eemian temperature reconstructions and its uncertainty range, an accurate TAC record is of high importance. The following example illustrates this importance: the NEEM-derived surface temperature anomaly (NEEM community members, 2013) at 126 ka is 7.5 ± 1.8 °C (relative to the last 1000 years) without accounting for elevation changes; including the elevation change based on TAC measurements, the temperature estimate becomes 8 ± 4 °C. This means that more than half of the uncertainty of this temperature estimate is related to the uncertainty of past surface elevation.

Despite the importance that melt can have for the interpretation of TAC and other variables of ice core records, the number of studies analyzing the frequency of melt layers in Greenland ice cores is limited (Alley and Koci, 1988; Alley and Anandakrishnan, 1995).

This study investigates regional climate simulations and observations at seven deep Greenland ice core sites — Camp Century, Dye-3, EGRIP, GRIP, GISP2, NEEM, and NGRIP. Additionally, an ice cap in the vicinity of the ice sheet is examined — the Agassiz ice cap, located in the northern Canadian Arctic. TAC is derived from regional climate and melt simulations at these locations of interest (Sec. 2). Furthermore, the simulated local temperature and melt is evaluated, and the impact on TAC is estimated and compared with ice core observations (Sec. 3 and 4). The results indicate that Greenland ice core records from warm periods, such as the Eemian interglacial period, might be more affected by surface melt than previously considered (Sec. 5).

## 2 Methodology

**Climate and surface mass balance (SMB) simulations**

This study uses climate and surface mass balance (SMB) based on two Eemian time slice simulations with a fast version of the Norwegian Earth System Model (NorESM1-F; Guo et al., 2018) representing (constant) 125 and 115 ka conditions and

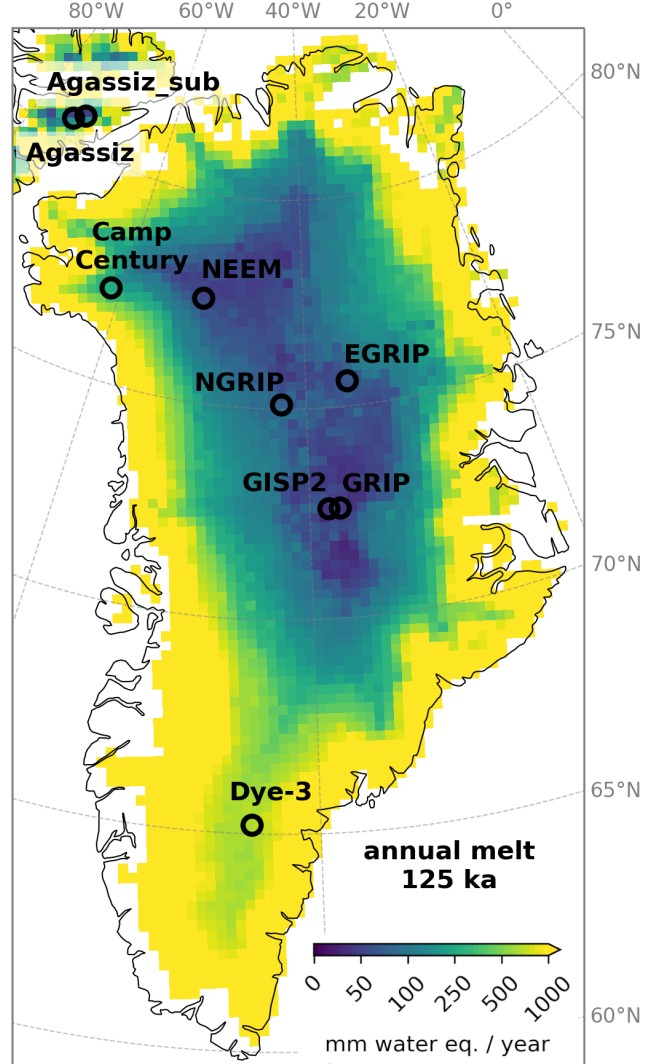

**Figure 1.** Overview map of Greenland ice core locations considered in this study. The gridded data shows the simulated annual melt rate under 125 ka conditions. Note: Agassiz_sub refers to a substitute location necessary due to the model topography misrepresentation (see Sec. 2).

one pre-industrial (PI; constant 1850 forcing) control simulation. These global simulations are dynamically downscaled over Greenland with the regional climate model Modèle Atmosphérique Régional (MAR, v3.6, 25 x 25 km), which was extensively validated over Greenland under present-day climate conditions (Fettweis, 2007; Fettweis et al., 2013a, 2017).

MAR employs a land surface model (SISVAT; Soil Ice Snow Vegetation Atmosphere Transfer) with a detailed snow energy balance (Gallée and Duynkerke, 1997) fully coupled to the model atmosphere. MAR's atmosphere uses the solar radiation scheme of (Morcrette et al., 2008) and accounts for the atmospheric hydrological cycle (including cloud microphysics) based

on Kessler (1969) and Lin et al. (1983). The snow-ice component of MAR is derived from the snowpack model Crocus (Brun et al., 1992) simulating mass and energy fluxes between snow layers, and reproducing snow grain properties as well as their effect on surface albedo. The MAR model has 24 atmospheric layers (up to 16 km above ground) and SISVAT 30 snowpack layers.

The NorESM-F experiments are spun up for 1000 years with constant 1850 forcing (greenhouse gas (GHG) concentrations and orbital parameters) to a quasi-equilibrium state. The PI simulation is run for another 1000 years with constant forcing. The two Eemian time slice simulations are branched off from the initial 1000-year spin-up and run for another 1000 years each with constant 125 ka and 115 ka forcing, respectively (changed GHG concentrations and orbital parameters compared to PI). For the MAR experiments, NorESM is run for another 30 years for each of the three experiments and the output is saved 6-hourly.

These 30 years are used as boundary forcing for MAR. After disregarding the first four years as spin-up, the final 26 years are used for the analysis (thin lines in Figs. 2, 3, 4, 5, 6, A1, A2, A3). All climate simulations use a fixed, modern ice sheet geometry, in the absence of a reliable Eemian ice sheet estimate and high computational costs of a coupling with an ice flow model (e.g., Clec'h et al., 2019).

The MAR SMB is analyzed in a study investigating the influence of climate model resolution and SMB model selection on
Eemian SMB simulations (Plach et al., 2018a) which amongst other things shows the high importance of solar insolation in Eemian simulations. Additionally, while providing the most complete representation of physical surface processes in the pool of investigated models, MAR shows a less negative SMB than an intermediate complexity model during the warmest Eemian simulations (mainly due to a higher ratio of refreezing).

Furthermore, the discussed SMB is used in a study investigating the Eemian Greenland ice sheet volume with a higher-order
ice sheet model (Plach et al., 2019). Plach et al. (2019) shows that different external SMB forcings show a larger influence on the Eemian ice volume minimum than sensitivity experiments performed with internal ice dynamics (like changed basal friction). The ice sheet simulations with the MAR SMB show a moderately smaller Eemian ice sheet with the difference equivalent to ~0.5 m of sea level rise (with respect to the modern ice sheet).

In this study, the MAR SMB simulations are analyzed at seven deep Greenland ice core locations — Camp Century, Dye-3,
EGRIP, GRIP, GISP2, NEEM, NGRIP — and an adjacent ice cap — the Agassiz ice cap (Fig. 1). Due to model topography misrepresentation at the ice sheet margins, i.e., the model topography is lower than in reality at the Agassiz ice cap location (model resolution 25 km), a substitute location (Agassiz_sub) in the vicinity of the ice cap, with a model elevation similar to the observed elevations, is chosen (Tab. 1).

**Observed surface melt**

The PI climate and SMB simulations are compared to present-day satellite and temperature observations at the locations of interest. The two observational melt day data sets are both derived from satellite-borne passive microwave radiometers — Scanning Multichannel Microwave Radiometer (SMMR), the Special Sensor Microwave/Imager (SSM/I), and the Special Sensor Microwave Imager/Sounder (SSMIS). The first data set, *MEaSUREs (Greenland Surface Melt Daily 25km EASE-Grid 2.0, Version 1)*, covers the years 1979 to 2012 and is available for the entire Northern Hemisphere. The melt onset is identified

**Table 1.** Greenland ice core locations.

| location | latitude (°N) | longitude (°W) | observed elevation (m) | model elevation (m) | model accumulation (m w.e./yr) PI \| 115 ka \| 125 ka |
|---|---|---|---|---|---|
| Agassiz | 80.7 | 73.1 | 1730 | 1575 | 0.22 \| 0.18 \| 0.26 |
| Agassiz_sub | 80.5 | 74.5 | 1730 | 1741 | 0.29 \| 0.24 \| 0.34 |
| Camp Century | 77.2 | 61.1 | 1890 | 1849 | 0.63 \| 0.52 \| 0.76 |
| Dye-3 | 65.2 | 43.8 | 2490 | 2444 | 0.65 \| 0.61 \| 0.74 |
| EGRIP | 75.6 | 36.0 | 2710 | 2684 | 0.13 \| 0.11 \| 0.14 |
| GISP2 | 72.6 | 38.5 | 3200 | 3198 | 0.20 \| 0.18 \| 0.22 |
| GRIP | 72.6 | 37.6 | 3230 | 3221 | 0.19 \| 0.18 \| 0.21 |
| NGRIP | 75.1 | 42.3 | 2920 | 2906 | 0.18 \| 0.16 \| 0.22 |
| NEEM | 77.5 | 51.0 | 2450 | 2429 | 0.26 \| 0.23 \| 0.34 |

Agassiz_sub refers to a substitute location used due model topography misrepresentation. Details see Sec. 2

by comparing 37 GHz, horizontally polarized (37 GHz H-Pol) brightness temperatures with dynamic thresholds associated with a melting snowpack (Mote, 2014). Unfortunately, the Agassiz ice cap is not covered by this data set. The second data set, T19H$_{melt}$, covers the whole MAR grid at 25 km from May to September for most years between 1979 and 2010. It uses data collected at K-band horizontal polarization (T19H) with a constant brightness temperature threshold of 227.5 K (Fettweis et al., 2011). Both satellite data sets are discussed to show their different sensitivities and to illustrate the uncertainty of these satellite-based melt observations.

The seasonal temperature observations at weather stations and 10 m borehole temperatures (representing annual mean temperatures from 1890 to 2014) are taken from a collection of shallow ice core records and weather station data (Faber, 2016). Finally, the bore hole temperatures from the Agassiz ice cap are taken from Vinther et al. (2008).

**Observed total air content (TAC)**

Firstly, the Dye-3 TAC for the ice core depth range of ~240 to 1920 m was extracted from Herron and Langway (1987, Fig. 4 therein). Since Souchez et al. (1998) indicate that ice from warmer periods (higher $\delta O^{18}$ values), likely Eemian, is located below 2000 m at Dye-3, the presented Dye-3 TAC record does not represent Eemian conditions. Secondly, the GRIP TAC data set (Raynaud, 1999) covers depths from ~120 to 2300 m and ~2780 to 2909 m, while an age mode is only provided for the upper part (oldest ice 41 ka). For the deeper sections of the core, a published unfolding of the GRIP core (Landais et al., 2003, age bands in Fig. 3 therein) is used to assign an age to the observations. Thirdly, the GISP2 TAC data was extracted from a supplement table of Yau et al. (2016) and covers the period from 127.6 to 115.4 ka. Fourthly, the NEEM TAC observations (NEEM community members, 2013) cover the deepest section of the NEEM ice core from ~2200 to 2500 m depth (corresponding to an age of ~75 to 128 ka; not continuous) and an example for Holocene conditions from depths between ~100 to 1400 m (no

age provided). Finally, the NGRIP TAC record (Eicher et al., 2016) includes the entire core from ~130 to 3080 m, however the sampling resolution varies. An age model is provided for the entire data set with a maximum age of ~120 ka. Note that only the Eemian sections for GRIP, GISP2, NEEM, and NGRIP are shown in Fig. 7.

**Calculation of the model-derived total air content (TAC)**

The model-derived TAC is calculated with the annual mean surface pressure and the annual mean near-surface temperature from the MAR regional climate simulations at every location of interest (Martinerie et al., 1992; Raynaud et al., 1997):

$$TAC = V_c \frac{P_c}{T_c} \frac{T_0}{P_0} \tag{1}$$

where $V_c$ is the pore volume at close-off in $cm^3/g$ of ice, $P_c$ the mean atmospheric pressure at the elevation of the close-off depth interval in $mbar$, $T_c$ the firn temperature prevailing at the same depth interval in $K$, $P_0$ the standard pressure (1013 $mbar$), and $T_0$ the standard temperature (273 $K$). $V_c$ is calculated as a function of $T_c$ following an empirical relation (Martinerie et al., 1994; Raynaud et al., 1997):

$$V_c = (6.95 \times 10^{-4} T_c) - 0.043 \tag{2}$$

This theoretical TAC is then reduced ($TAC_{red}$) depending on the percentage of refreezing of the annual accumulation ($RZ_{per}$):

$$TAC_{red} = TAC \times \left(1 - \frac{RZ_{per}}{100}\right) + TAC_{refrozen} \times \left(\frac{RZ_{per}}{100}\right) \tag{3}$$

where $TAC_{refrozen}$ is calculated using Henry's solubility law (Sander, 2015) for $N_2$ and $O_2$ (neglecting other atmospheric gases) to account for air that is dissolved in the meltwater before refreezing:

$$TAC_{refrozen} = C_{a,N2} + C_{a,O2}, \tag{4}$$

with $C_{a,N2}$, and $C_{a,O2}$ being the aqueous-phase concentration of $N_2$ and $O_2$, respectively:

$$C_{a,N2} = P_c * C_{atm,N2} * H^{cp,N2} \tag{5}$$

and

$$C_{a,O2} = P_c * C_{atm,O2} * H^{cp,O2} \tag{6}$$

where $C_{atm,N2}$ and $C_{atm,O2}$ are the atmospheric concentration ratios (0.79 and 0.21) and $H^{cp,N2}$, $H^{cp,O2}$ are Henry's solubility constants ($10.49 \times 10^{-6}$ and $2.1982 \times 10^{-5}$) for $N_2$ and $O_2$, respectively. Henry's law assumes that the meltwater is in equilibrium with the ambient air at a temperature of 273 K and at the local atmospheric pressure (Eqs. 5 and 6). No air is occluded in the form of bubbles in the freezing process.

## 3 Results

**Temperatures**

The simulated PI annual mean (near-surface) temperatures (1850 climate forcing) at the eight locations of interest (Fig. 2; black columns; short bold lines - ensemble means; short thin lines - individual model years) generally fit well with observations from weather stations (Fig. 2; long bold lines in black; standard deviation in gray shading). However, the annual means inferred from 10 m borehole temperatures (Fig. 2; long bold lines in gray; average of the years 1980 to 2014) are consistently colder than the simulated PI means. The lower borehole temperatures represent snow temperatures which are typically cooler than the ambient air temperatures. Only at the Agassiz site, the borehole temperatures are higher. This exception is likely related to the usage of a substitute location (see Sec. 2).

The annual mean temperatures at most locations only vary by 0.5 °C between the time slice simulations, i.e., no large difference between PI (Fig. 2; black) and warmest Eemian simulations (Fig. 2; orange). This is to be excepted since the annually integrated solar irradiance is similar in all time slices.

However, the varying Eemian seasonality (Yin and Berger, 2010) results in consistently ~3-4 °C (with respect to PI; black) warmer summer (JJA; June-July-August) temperatures at all locations for mid Eemian conditions (125 ka: orange) and cooler temperatures for late Eemian conditions (115 ka: blue). The simulated PI summer temperatures (Fig. 3; black columns; short bold lines - ensemble means; short thin lines - individual model years) show good agreement with observations from weather stations (Fig. 3, long bold lines in black).

The precipitation-weighted temperatures (Fig. A1) show a similar pattern as the JJA temperatures (Fig. 3). This is understandable since most precipitation in Greenland falls around the summer month and these temperatures are calculated by multiplying daily temperatures with daily precipitation, summing up the results over the year and then dividing by the sum of the annual precipitation, i.e., precipitation is used as a weight, instead of time in annual mean temperatures. Precipitation-weighted temperatures are arguably closer to what is recorded in an ice core (temperature at the time of deposition) and these temperatures show a less pronounced warming for mid Eemian conditions (125 ka: orange), i.e., maximum 3 °C warmer compared to PI (black).

**Number of melt days**

Passive microwave satellite data shows a strong difference in observed melt days per year (presence of surface water) (Fig. 4; first three columns from the left; brown and green) between central ice core locations (GRIP, GISP2, NGRIP, NEEM, EGRIP),

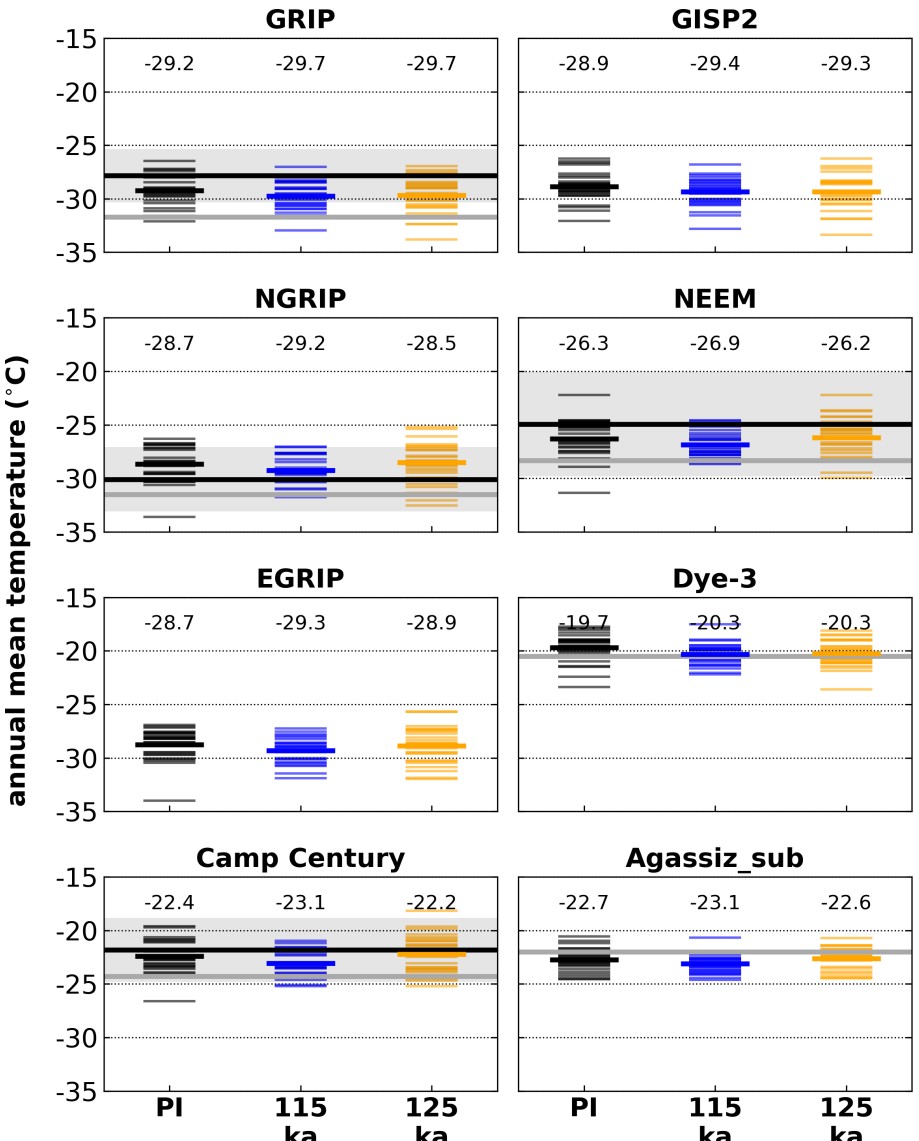

**Figure 2.** Annual mean (near-surface) temperature at Greenland ice core locations simulated by the climate model MAR for three time slices. Individual model years (short thin lines) and their mean (short bold lines, numerical values on top of columns) are compared to mean observations from weather stations (long bold lines in black), their corresponding standard deviation (gray shading), and 10 m borehole temperatures (annual mean; long bold lines in gray).

where surface melt is sparse, and locations closer to the margins (Camp Century, Dye-3) and ice caps (Agassiz), where melt is much more frequent. Central locations show between 0 and ~1 melt days year$^{-1}$ in the last ~30 years for which satellite data is available. The exact values vary depending on the location, satellite data set, and whether the extreme melt event of 2012 is included.

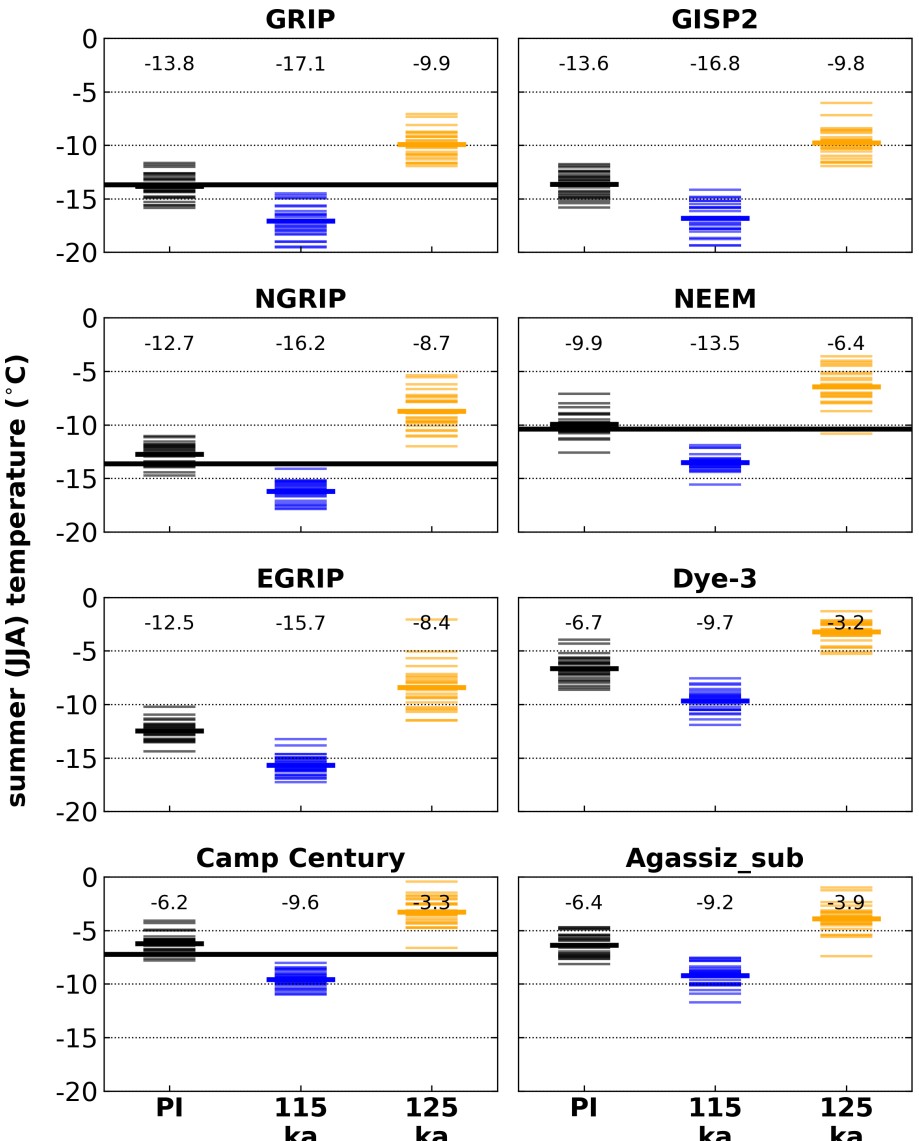

**Figure 3.** Mean (near-surface) JJA (June-July-August) temperature at Greenland ice core locations simulated by the climate model MAR for three time slices. Individual model years (short thin lines) and the mean (short bold lines, numerical value on top of columns) are compared to mean observations from weather stations (long bold lines in black).

The simulated PI melt day frequency (Fig. 4, black columns) shows good agreement with the observations (Fig. 4; brown and green columns), i.e., low melt frequencies at the central locations and higher melt frequencies at locations at the margins. However, the simulated PI melt frequencies are generally lower than present-day observations (especially at the Agassiz location), with the exception of Dye-3 which shows a higher simulated melt frequency.

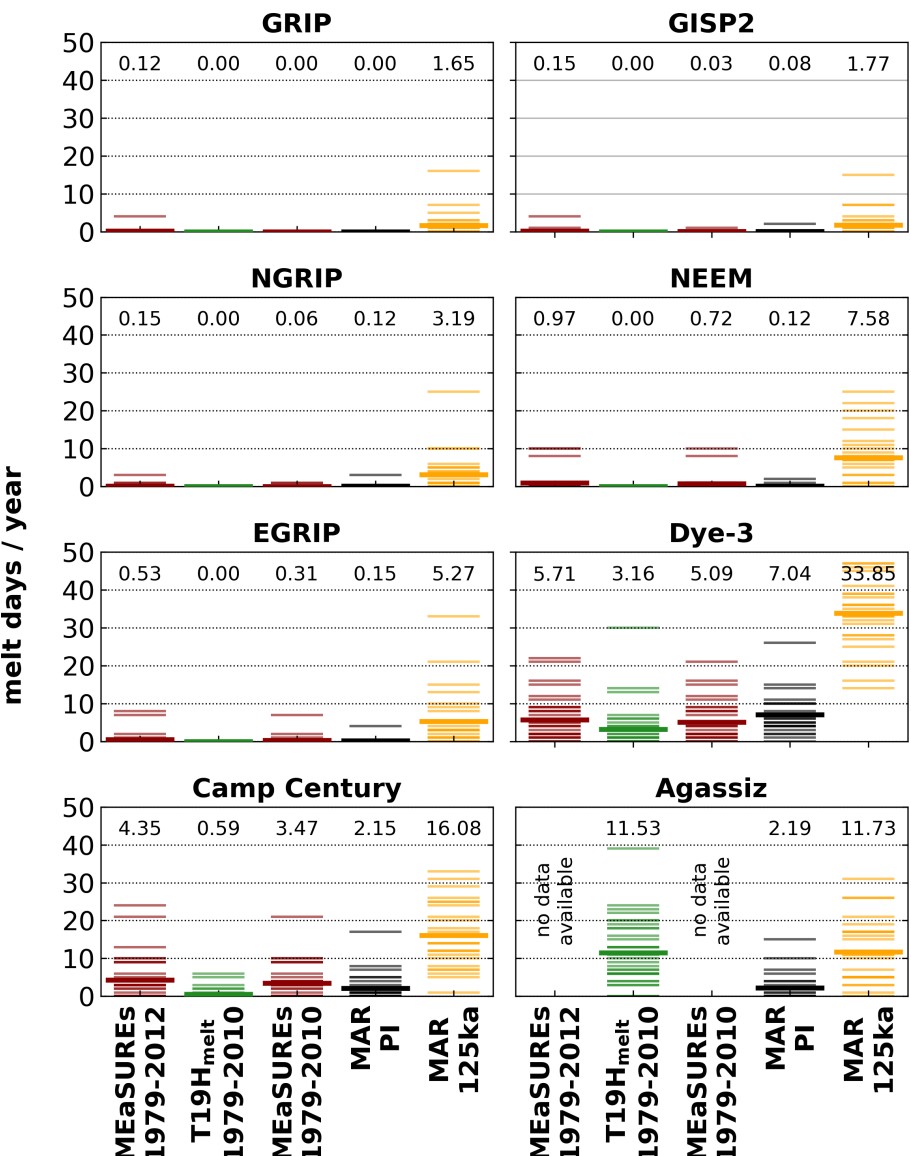

**Figure 4.** Annual melt days at Greenland ice core locations derived from satellite data and simulated by the climate model MAR. Observations in the first three columns from the left are compared with simulations in the fourth and fifth column. Columns from the left: (1) Passive microwave data from MEaSUREs (1979 to 2012); (2) The same data as in (1) but with a different processing (T19H$_{melt}$; Fettweis et al., 2011) (1979 to 2010); (3) the MEaSUREs data set excluding the extreme melt year 2012 (1979 to 2010); (4) Simulated melt for pre-industrial (PI) and (5) 125 ka conditions. Individual model years (thin lines) and the ensemble means (bold lines, numerical values on top of columns) are shown. For Agassiz, simulation results for the substitute location are shown; as discussed in Sec. 2.

**Melt and refreezing**

The 125 ka simulations (Fig. 4; orange columns) show a significantly higher melt frequency at all locations (more than 30 melt days year$^{-1}$ at Dye-3), compared to the PI simulations (Fig. 4; black columns) and observations (Fig. 4; brown/green

columns). The SMB simulations show surface melt at all ice core locations during the warm mid Eemian with an annual melt water production (Fig. A2) for warmer locations of ~300 mm w.e. year$^{-1}$ (Camp Century) and ~600 mm w.e. year$^{-1}$ (Dye-3). However, even modern dry, high altitude locations show an annual surface melt of ~60 (GRIP, GISP2), 80 (NGRIP) and up to 120 mm w.e. year$^{-1}$ (EGRIP). NEEM shows ~150 mm w.e. year$^{-1}$ for the warmest Eemian simulations.

The mean simulated amount of refreezing exceeds 40 % of the annual accumulation at most ice core locations under warm mid Eemian conditions (Fig. 5; thick orange lines). Even at the highest locations, GRIP and GISP2 at ~3200 m elevation, refreezing surpasses 25% of the annual accumulation under 125 ka conditions. The largest amount of refreezing is simulated at Agassiz_sub, EGRIP, and Dye-3 where refreezing percentages reach 80 to 90%.

**Total air content (TAC)**

Theoretical TAC derived from simulated surface pressure and annual mean temperature (Raynaud et al., 1997) and reduced according to the amount of simulated refreezing (Fig. 6 and Sec. 2) shows significantly lower values for the 125 ka simulations. Most of the higher ice core locations (GRIP, GISP2, NGRIP, NEEM, EGRIP and Camp Century) show simulated TAC values between 45 and 70 ml kg$^{-1}$ on average, whereas the respective PI values are between 90 and 100 ml kg$^{-1}$. At Dye-3 the simulated TAC is about 25 ml kg$^{-1}$ on average for the warm 125 ka Eemian simulations compared to 75 ml kg$^{-1}$ during PI. Observed Holocene TAC from ice core records (Fig. 6; horizontal gray shading) fit well with the PI simulations, while observed Eemian TAC (Fig. 6; horizontal orange shading) is not as low as the simulated values.

The Eemian ranges in Fig. 6 are calculated as the average (plus/minus two standard deviations) of the lowest 10 % of observed Eemian TAC (Fig. 7; used observations are indicated in orange) for NEEM and NGRIP. Due to the low number of Eemian observations at GRIP and GISP2, a different threshold of 20 % is used for this core. For the calculation of the late Holocene ranges in Fig. 6, observations younger than 1000, 2000, and 4000 years, are used for GRIP, Dye-3, and NGRIP, respectively. The late Holocene range for NEEM is calculated from the entire Holocene example provided in the NEEM community members (2013) data (nine data points; no age provided).

Finally, TAC observations from the deeper ice core sections (i.e., possibly Eemian; Fig. 7; NEEM, GRIP, GISP2, NGRIP; circles; inverted y-axes) are compared with mean simulated TAC for 115 ka (Fig. 7; blue line) and 125 ka conditions (Fig. 7; orange line). For Dye-3 the entire TAC record is shown due to the lack of Eemian observations. However, the ice at the bottom of Dye-3 has been shown to contain pre-Eemian ice (Willerslev et al., 2007). Note that NEEM and GRIP are shown against age based on a more robust chronology involving "unfolding the ice" (NEEM community members, 2013; Landais et al., 2003), while NGRIP and Dye-3 are shown against core depth.

The 115 ka simulations generally fit well with the late Eemian (NEEM, GRIP, GISP2, NGRIP) and Holocene (Dye-3) observations, while the 125 ka simulations are lower than the observations. For NEEM, the lowest TAC observations are within the ice core section influenced by melt (gray shading in Fig. 7; NEEM community members, 2013).

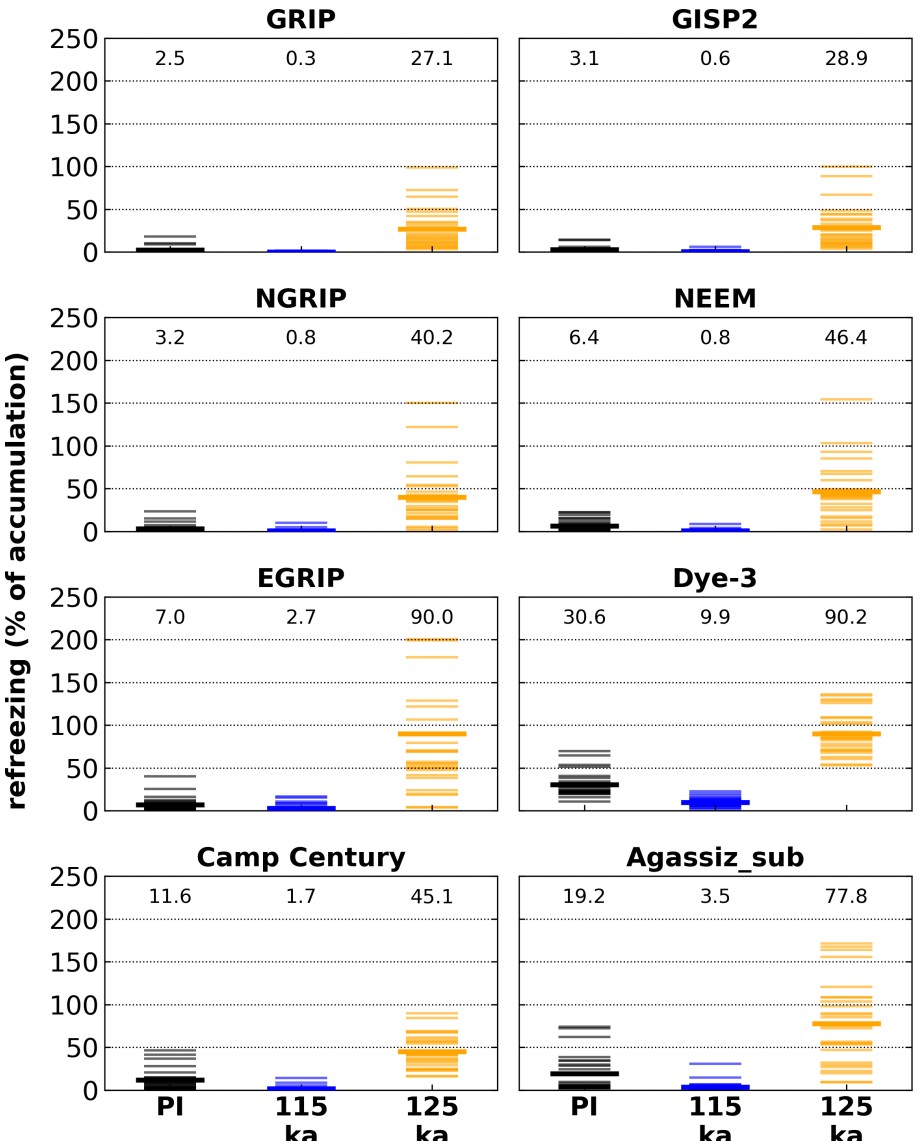

**Figure 5.** Annual refreezing percentage (of accumulation) at Greenland ice core locations simulated by the climate model MAR for three time slices. Individual model year percentages (thin lines) and the simulation ensemble mean percentages (bold lines, numerical values on top of columns) are shown.

## 4   Discussion

The enhanced Eemian seasonality (Yin and Berger, 2010) and warmer Eemian summers (CAPE Last Interglacial Project Members, 2006; Otto-Bliesner et al., 2013; Capron et al., 2014) are indicators of elevated melt during this period. The recent extreme melt event in Greenland in 2012 and a similar event in 1889 (Nghiem et al., 2012) demonstrate that surface melt on

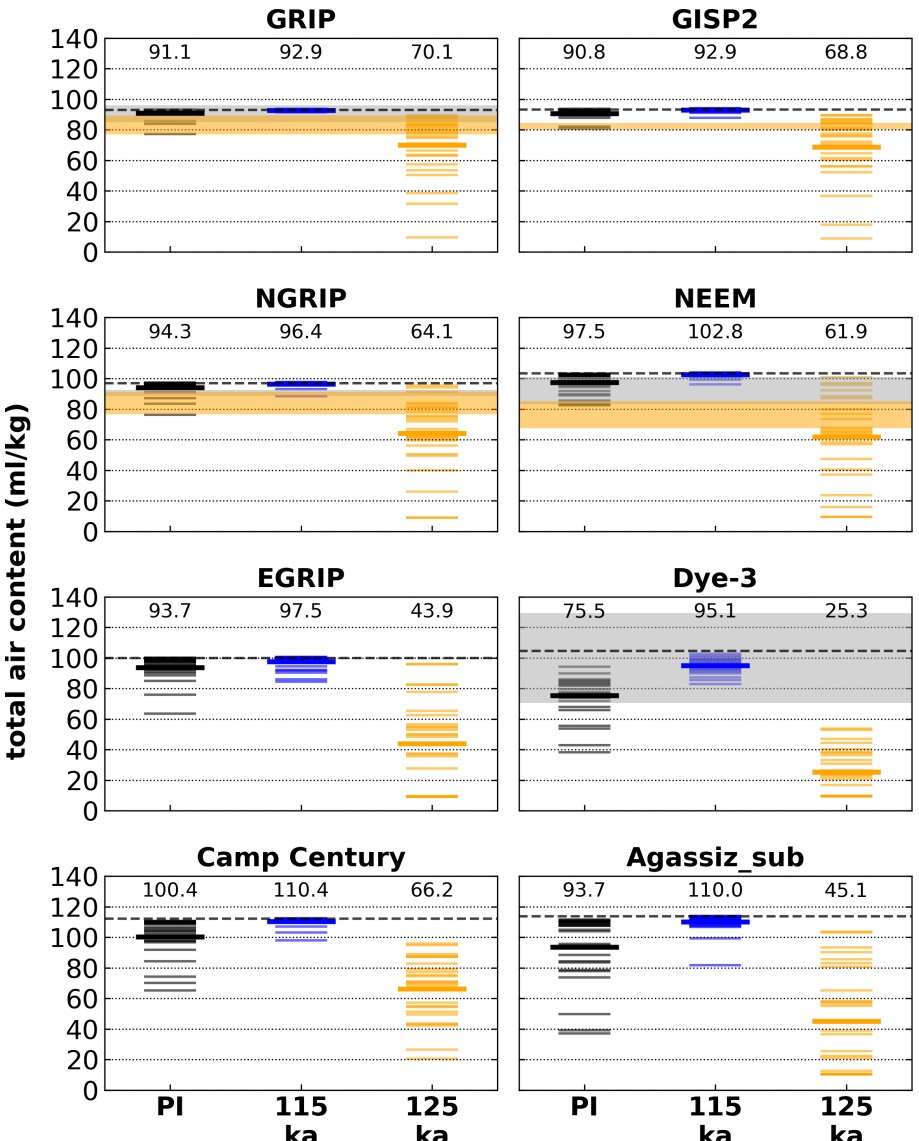

**Figure 6.** Calculated total air content (TAC) at Greenland ice core locations derived from simulations with the climate model MAR for three time slices (see method in Sec. 2). Individual model years (thin lines) and the simulation ensemble means (bold lines, numerical values on top of columns) are compared to observed late Holocene and Eemian ranges (horizontal gray and orange shading, respectively; two standard deviations). Dashed lines illustrate the model-derived TAC before reducing it by the refreezing percentage (not distinguishable for the respective time slices; see Sec. 2). Note: The Holocene range at NGRIP is very narrow and almost completely overlaps with the Eemian range and there is no Holocene range for GISP2 and no Eemian range for Dye-3.

the entire Greenland ice sheet, even at the summit of Greenland, is possible under recent climate conditions. Even though these extreme Greenland-wide melt events were caused by a rare large-scale atmospheric pattern (Neff et al., 2014) and were further

enhanced by an externally caused albedo lowering (ash deposition from forest fires; Keegan et al., 2014), it is likely that such events are more frequent in a warmer climate such as the Eemian interglacial period.

The simulations discussed in this study (regional climate plus a full surface energy balance) indicate surface melt and refreezing (Fig. 4 and 5) at all deep Greenland ice core locations. Even central Greenland locations close to Summit (GRIP, GISP2) show a melt of ~60 mm year$^{-1}$ (Fig. A2). Due to this high surface melt, TAC derived from these simulations are

between ~25 % (GRIP, GISP2) and ~80 % (Dye-3, EGRIP) lower than modern (PI) values (Fig. 6). Even though the presented climate simulations show such extensive melt, there are several reasons why these simulations can be interpreted as conservative estimates: (1) The simulated PI melt frequency is mostly lower than satellite observations (Fig. 4; black versus brown/green columns). However, the observation of higher melt frequencies can likely also be related to the effects of recent global warming which are not represented in the PI climate simulations. (2) Processes like ash deposition which were partly responsible for

the extreme Greenland melt events of 2012 and 1889 (Keegan et al., 2014) are not simulated. (3) The climate simulations use a fixed, modern ice sheet geometry and including the neglected lowering and retreat of the Eemian ice sheet would likely increase the simulate warming in many regions.

Many studies suggest a substantial Eemian ice volume reduction (e.g, Van de Berg et al., 2011) particularly in the marginal regions — an overview of previous Eemian studies can be found in Plach et al. (2018a). The use of a fixed ice sheet undoubtedly

adds additional uncertainties to the presented melt simulations — e.g., neglecting modifications of local wind patterns and surface albedo as regions become deglaciated impacting local near-surface temperature (Merz et al., 2014a), local orographic precipitation following the slopes of the ice sheet (Merz et al., 2014b), or increased katabatic winds caused by steeper ice sheet slopes (Gallée and Pettré, 1998; Clec'h et al., 2019). However, these uncertainties are much stronger in marginal than in high altitude regions where the ice elevation changes were more limited. After all, a future, more exhaustive evaluation of Eemian

melt at the ice cores sites should investigate different possible ice sheet geometries.

Furthermore, the absence of a simulated annual warming, and proxy data showing Eemian peak temperatures as high as +7.5 ± 1.8 °C (NEEM community members, 2013, without altitude corrections) and +8.5 ± 2.5 °C (Landais et al., 2016) for NEEM (the North Greenland Eemian Ice Drilling project in northwest Greenland), and +5.2 ± 2.3 °C (Landais et al., 2016, lower bound as the record only starts after the peak Eemian warming) for NGRIP (North Greenland Ice Core Project) indicate

that the climate simulations might include a cold bias. The simulated JJA temperatures (Fig. 3) and the simulated precipitation-weighted temperatures (Fig. A1) show a peak warming of only ~3-4 °C and ~3 °C, respectively. However, the fact that NEEM community members (2013) infer an elevation (at the deposition site) of several hundred metres higher than at NEEM today complicates the interpretation of how well the simulated temperatures fit the proxy-derived observations.

Focusing again on the comparison of melt observations and simulations (Fig. 4), a strong underestimation of melt at the

Agassiz site in the PI simulations becomes apparent. This strong underestimation is likely related to the use of a substitute location (geographically shifted, with similar model and observed elevation) necessary due to low model topography at the original core site causing unrealistically high melt simulations. Furthermore, the Agassiz site is only covered by the satellite data set which appears to be less sensitive to melt (T19H$_{\mathrm{melt}}$ less melt than MEaSUREs at all sites) and although Eemian ice is absent at the Agassiz site, the simulated Eemian refreezing percentage (Fig. 5) of approximately 80% is consistent with

the Agassiz melt record which indicates a complete melt of the annual accumulation during the Holocene optimum ~10-11 ka (Fisher et al., 2012; Lecavalier et al., 2017).

Another important aspect for the melt interpretation is the formation of melt layers and the amount of meltwater needed to form a (visible) melt layer. While the presented TAC calculations assume Henry's solubility law (Sander, 2015) for the air content of the melt layer, the formation of a melt layer in an ice core is a complicated process, e.g., depending on prevailing
snow properties. A higher number of melt layers is not just the result of uniformly higher summer temperatures, but a combination of an increased contrast between the pre-melt snow pack temperatures (strongly influenced by winter temperature) and the summer melt rate (a function of summer temperature) (Pfeffer and Humphrey, 1998). Therefore, the enhanced Eemian seasonality might have been favourable for the formation of melt layers.

The simulated 125 ka TACs are consistently lower than the observations (Fig. 6 and 7). However, at NEEM — the ice core
with the most complete Eemian record (likely including peak warming) — the simulated 125 ka TAC seem to be most similar to the lowest observations, indicating that the high amount of simulated melt could explain these observations. The variability of the observed NEEM TAC in the suggested melt zone between 127 and 118.3 ka (gray shading;  NEEM community members, 2013) is large, likely due to the varying influence of melt layers.

The Eemian TAC measurements at GRIP, GISP2, and NGRIP also show reduced values (not as low as at NEEM), which can
be interpreted in a similar way as at NEEM — GRIP, GISP2, and NGRIP might have been influenced by Eemian melt as well. The simulated 125 ka TAC for all three locations are strongly reduced (relative to PI levels), but do not reach levels as low as at NEEM. However, these reduced TAC levels could indicate significant surface melt.

Overall the lack of a better agreement between observed and simulated Eemian TAC (i.e., few TAC observations as low as the simulations) could be related to the sparse number of Eemian peak warming observations (most ice core records only
start after the peak warming; particularly at GRIP, GISP2, NGRIP, and Dye-3). However, another possible explanation could be a shift of the precipitation rates in central Greenland towards much higher values during the Eemian interglacial period. Unfortunately, accumulation rates are unconstrained for the Eemian sections of Greenland ice cores.

Furthermore, another uncertainty to the interpretation of the simulations is the effect of the higher Eemian summer insolation on the TAC. An anti-correlation between local summer insolation and TAC is known in ice core records from East Antarctica
during the last 400000 years (Raynaud et al., 2007) and the insolation signal is also found in Greenlandic TAC (NGRIP, Eicher et al., 2016). NEEM community members (2013) estimate (based on data from the Holocene optimum) that the summer insolation could account for 50% of the observed Eemain TAC changes at NEEM.

Nevertheless, the possibility of a melt-induced reduction of TAC should be considered for the interpretation of Eemian air content to estimate ice surface elevation changes. An early interpretation of the first Greenland ice cores (Camp Century, Dye-
3) suggested an extreme scenario for Eemian Greenland with extensive melt and a much smaller ice sheet leading to a sea level rise of 6 m (Koerner, 1989). However, this scenario was rejected by later ice core studies showing evidence of Eemian ice (especially NGRIP and NEEM; North Greenland Ice Core Project members et al., 2004; NEEM community members, 2013). Furthermore, GRIP TAC measurements (Raynaud, 1999) have been interpreted as evidence for the elevation of the summit sites having remained above 3000 m of altitude during the Eemian and GRIP deuterium excess measurements remain in the

normal range during the Eemian (Landais et al., 2003). However, this last interpretation can be challenged by measurements of a NEEM Holocene melt layer, suggesting that the melt layer mainly influences TAC and $CH_4$ observations, while other variables like deuterium excess may be less influenced by melt (NEEM community members, 2013).

The climate simulations show surface melt at all deep ice core locations and at the Agassiz ice cap under 125 ka climate conditions (Fig. 4 and A2; orange column). Even locations near the summit of Greenland (GRIP, GISP2, and NGRIP) show a few melt days year$^{-1}$ on average (defined as >8 mm day$^{-1}$) during these warm Eemian simulations. NEEM, the ice core location with the longest Eemian record, shows ~8 melt days year$^{-1}$. While the presence of Eemian surface melt at NEEM was acknowledged previously (NEEM community members, 2013), the lower TAC observations at GRIP, GISP2, and NGRIP could as well be related to Eemian surface melt, rather than stable or higher elevations.

Finally, it should be emphasized that a robust estimate of Eemian Greenland surface melt is challenging to obtain with a single climate model. Ideally there should be an ensemble of climate models to explore model biases and uncertainties. However, as pointed out earlier in this discussion, there are several reasons why the presented climate simulations could be on the lower end of available climate model in terms of the amount of simulated Eemian melt. It is likely that there are other climate models which show more extensive Eemian surface melt.

In the future, an analysis of individual or ensemble Eemian climate simulations would benefit from a comparison of the observed extreme melt event in 2012 (and similar events in the recent past) with simulated extreme Eemian melt events. Relationships in the Eemian simulations between air temperature and local wind patterns, and the simulated melt could be analyzed and used to identify specific weather patterns leading to high surface melt in the simulations (e.g. similar analysis performed by Neff et al. (2014); Keegan et al. (2014); Fettweis et al. (2013b); Tedesco and Fettweis (2020)).

## 5 Conclusions

Using regional climate simulations (including a full surface energy balance) this study shows surface melt at all Greenland ice core locations during the Eemian interglacial period (e.g., GRIP, GISP2: ~60 mm w.e. year$^{-1}$; NGRIP: ~150 mm w.e. year$^{-1}$). The amount of refreezing exceeds 25 % of the annual accumulation at the summit of Greenland (GRIP, GISP2) and reaches values as high as 90 % at less central locations like Dye-3 and EGRIP. The simulated air pressure, temperature, and refreezing are used to estimate Eemian total air content (TAC) and high melt rates could explain the low corresponding ice core TAC observations. This is true even though the discussed simulations could show conservative melt estimates (several potentially melt-increasing processes are neglected). Therefore, the possibility of widespread surface melt should be considered for the interpretation of Greenlandic total air content records (as an elevation proxy) from warm periods such as the Eemian interglacial period. Finally, a robust map of Eemian melt estimates in Greenland in combination with accumulation patterns could be used to identify potential future ice cores sites on Greenland. Such a procedure would increase the chances of finding Eemian ice influenced by a minimum amount of melt layers. These sites will have relatively high accumulation combined with low surface melt.

## 6 Code availability

The MAR code is available at: http://mar.cnrs.fr (last access: 27.11.2020)

## 7 Data availability

The Eemian MAR simulations are available from the corresponding author upon request. *MEaSUREs Greenland Surface Melt Daily 25km EASE-Grid 2.0, Version 1* (Mote, 2014) is freely available at: https://nsidc.org/data/nsidc-0533/versions/1 (last access: 27.11.2020). For more information and to request the T19H$_{melt}$ data (Fettweis et al., 2011) please contact Xavier Fettweis (xavier.fettweis@uliege.be). For more information and to request the collection of Greenland shallow ice core and weather station data (Faber, 2016) please contact Anne-Katrine Faber (anne-katrine.faber@uib.no). The TAC observations

at NEEM (NEEM community members, 2013) are freely available at: http://www.iceandclimate.nbi.ku.dk/data/ (last access: 27.11.2020). The GRIP TAC (Raynaud, 1999) is freely available at: https://doi.pangaea.de/10.1594/PANGAEA.55086 (last access: 27.11.2020). The GISP2 TAC is freely available as a supplement to Yau et al. (2016). The NGRIP TAC (Eicher et al., 2016) is freely available at: https://www.ncdc.noaa.gov/paleo-search/study/20569 (last access: 27.11.2020). For more information and to request the Dye-3 data, please contact Sindhu Vudayagiri (sindhu.v@nbi.ku.dk) or Thomas Blunier (blu-

nier@nbi.ku.dk).

## Appendix A

*Author contributions.* AP and BMV designed the study with contributions from KHN. Dye-3 total air content data was extracted from Fig. 4 in Herron and Langway (1987) by SV. AP made the figures and wrote the text with input from BMV, KHN, SV, and TB.

*Competing interests.* The authors declare that they have no conflict of interest.

*Acknowledgements.* The research leading to these results has received funding from the European Research Council under the European Community's Seventh Framework Programme (FP7/2007-2013) / ERC grant agreement 610055 as part of the ice2ice project. We thank Chuncheng Guo for performing the Eemian NorESM simulations and Sébastien Le clec'h for downscaling the NorESM simulations with the regional model MAR. Furthermore, we would like to thank Anne-Katrine Faber for valuable discussions and providing the shallow ice core data she compiled during her PhD. We also thank Xavier Fettweis for providing the T19H$_{melt}$ data. Furthermore, we very much thank the

editor Eric Wolff and two anonymous referees for their comments and suggestions which significantly improved this manuscript.

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

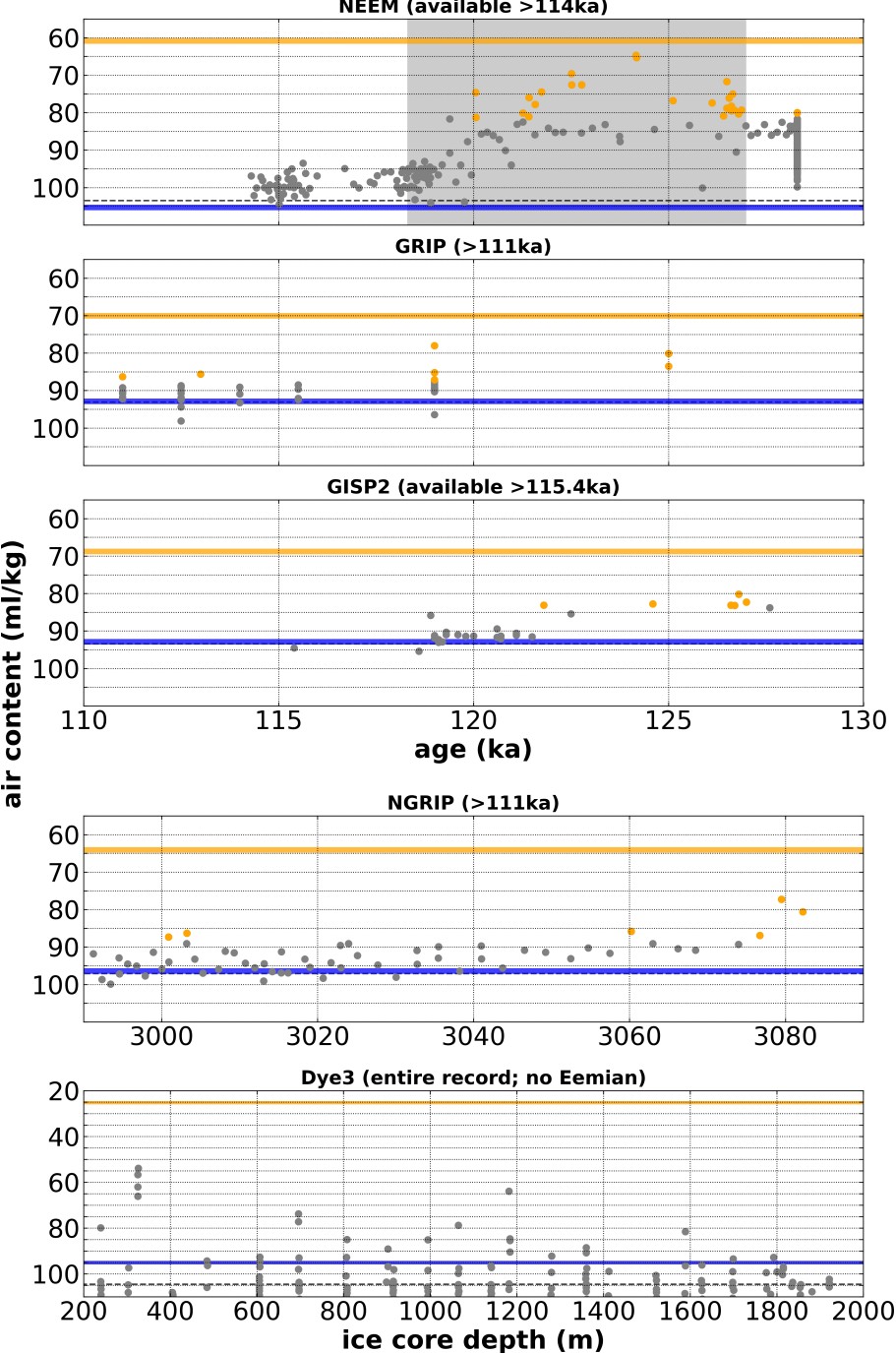

**Figure 7.** Observed TAC from five Greenland ice cores — NEEM, GRIP, GISP2, NGRIP, Dye-3. Observations (circles) are compared with mean simulated TAC for 115 ka (blue lines) and 125 ka simulations (orange lines). Furthermore, data points used to calculate the Eemian range in Fig. 6 (orange circles) and the model-derived TAC before reducing it by the refreezing percentage (dashed lines; see Sec. 2) are shown. Note: NEEM, GRIP, and GISP2 are shown against age (robust age models), while NGRIP and Dye-3 are shown against ice core depth. The NEEM melt zone (NEEM community members, 2013) is highlighted with a gray shading. The y-axes are reversed.

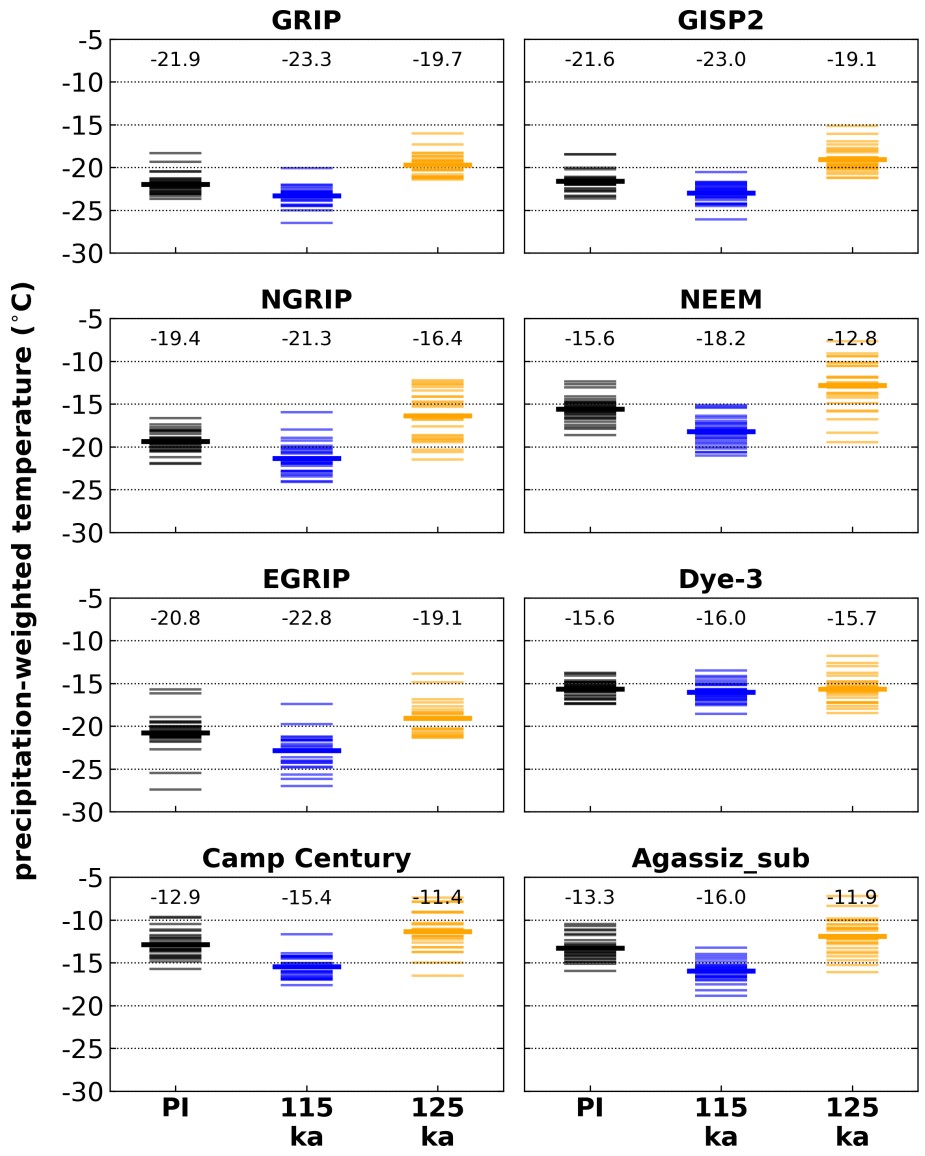

**Figure A1.** Annual mean precipitation-weighted temperature at Greenland ice core locations simulated by the climate model MAR for three time slices. Individual model years (thin lines) and the mean (bold lines, numerical values on top of columns) are shown.

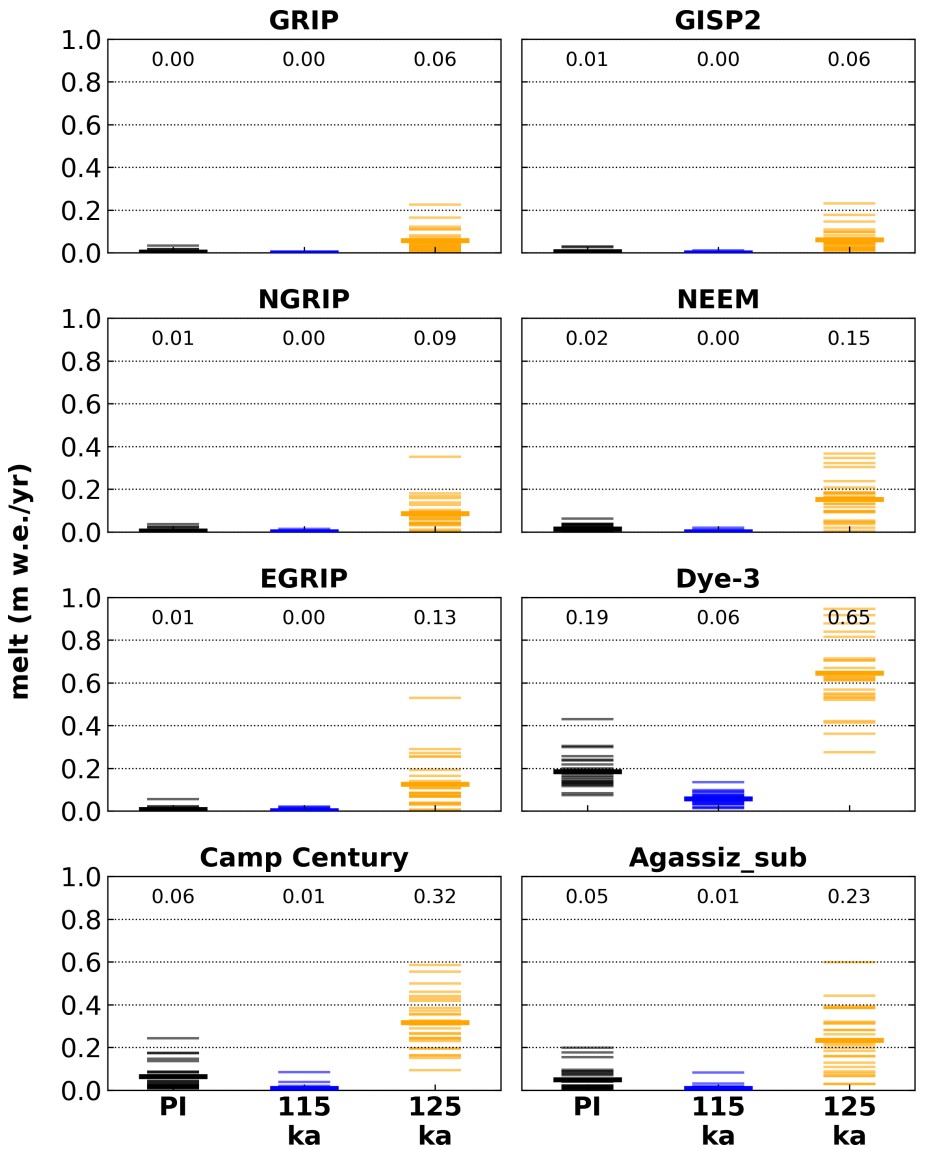

**Figure A2.** Annual melt at Greenland ice core locations simulated by the climate model MAR for three time slices. Individual model years (thin lines) and the mean (bold lines, numerical values on top of columns) are shown.

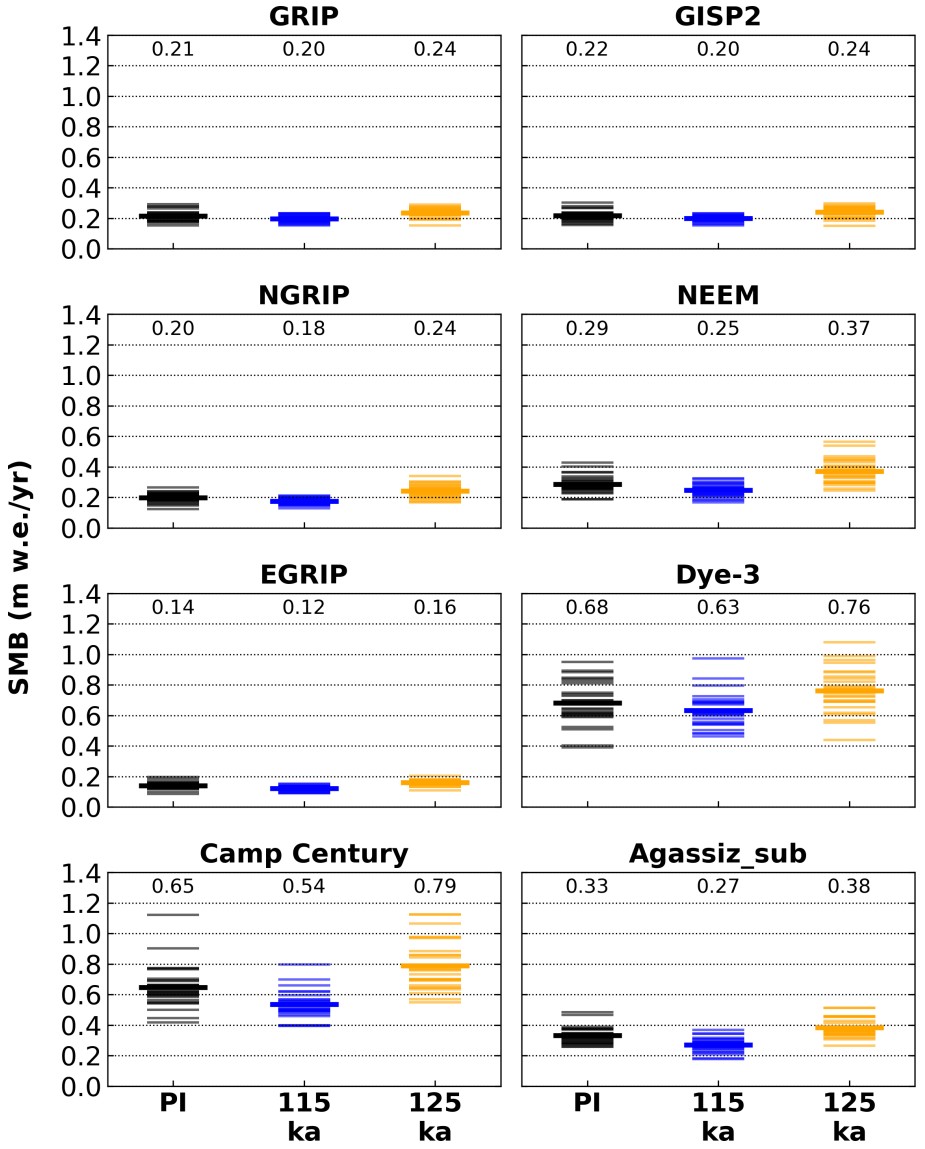

**Figure A3.** Annual Surface Mass Balance (SMB) at Greenland ice core locations simulated by the climate model MAR for three time slices. Individual model years (thin lines) and the mean (bold lines, numerical values on top of columns) are shown.