# Peer review of "Greenland climate simulations show high Eemian surface melt"

_Climate of the Past, 2020_

## Referee Comment (RC1) · Anonymous Referee #1 · 21 Sep 2020

The paper by Plach et al. addresses two topics of interest. First, it simulates the frequency of surface melt over Greenland during the Eemian from ∼130ka to 115ka. The authors use the surface mass balance output from the Norwegian Earth System Model with time slices at 125 and 115ka in combination with the Modele Atmospherique to attain regional climates over Greenland. To test the viability of their approach, the models were validated by comparing them to preindustrial conditions. The authors conclude from the preindustrial comparison that their model is likely conservative in estimating the percentage of melt. They find that at 125ka the percentage of melt could exceed 25% of the annual accumulation at GRIP and up to 90% in less central locations such as Dye-3. These are impressive results and could be important for understanding seasonality effects on surface melt and on ice sheet mass balance.

[Figure]

Secondly Plach et al. addresses how surface melt can affect ice core records, particularly TAC. The authors point out that the interpretation of TAC as a unique proxy for elevation is complicated by the influence of surface melt during warm periods especially the Eemian where these layers cannot be visually identified due to thinning. Melt layers are generally bubble free with the amount of air in the melt layer being primarily dictated by Henry's law. The result is that melt layers have significantly lower TAC. Plach et al. use the results of their melt simulations to calculate theoretical Eemian TAC values which they call TACred and then compare these values to published data. The results of the model derived TAC is consistently lower than the measured data at 125ka but similar at 115ka and the preindustrial.

General comments, suggestions and edits

Both the extent of melt over Greenland, and how melt can affect TAC during warm periods are important topics and of value to the paleoclimate community. The approach of the authors to account for melt in TAC is novel and could help explain some of the anonymously low TAC values we see during the Eemian in some ice core records. I find that the subject matter fits well within the scope of Climate of the Past. However, the paper could use substantial revision, as noted below.

First, I wonder if it would be better to split this into two papers, one on the Greenland surface melt simulations and a second on the potential effects of melt on TAC records and their interpretation. This would enable a more detailed analysis of the surface melt simulations and the interpretation of them, as well as a more robust comparison between the derived or simulated TAC and the measured data. On both the melt simulation results and the derived TAC, I would like to see a more robust interpretation of the results. Assuming the authors choose to keep this as a single paper the following are my suggestion.

Major Suggestions: Title: The title makes no mention of total air content but yet this is a major component of the paper. I suggest changing the title to include total air content.

Abstract: In contrast to the title the abstract is excessively focused on the TAC results. I suggest adding a few sentences about the melt simulations.

Ln 101-120: Calculation of the model-derived total air content (TAC) The derived TAC does not account for insolation. As mentioned earlier in the paper (Eicher et al. 2016) show that TAC at high accumulation sites such as in Greenland have a similar imprint from insolation as in Antarctic records. The implication is that the comparison shown in figure 6 is between TAC records with the effects of insolation and derived values without the effect of insolation. This will bias the results. While this discrepancy is included in the Discussion section, it would be preferable here. Note that (Eicher et al. 2016) provides an insolation sensitivity (5.7 *10-9 mL kg-1 J-1) for the integrated insolation threshold over 390Wm-2. This should provide the framework to approximate how much insolation affects the analysis in this paper.

Ln 164-170: Results: Total air content (TAC) and Figure 7 In addition to comparing the derived results with data, it would be useful to compare the derived results with and without TACred included. This would help determine the magnitude of the effect on TAC. While the derived TAC may be lower than the measured TAC for a number of reasons including insolation effects and elevation uncertainty, the difference between derived TAC and TACred may still be fairly accurate and informative.

Ln 95-96: and Figures 6 and 7 It is not clear to me why only the lowest 10% of TAC values (20% for GRIP) were used to determine the Eemian measured values for comparison to the derived values at the 125ka time slice. For the derived TAC the melt effect should be applicable to all TAC samples not just the ones with the lowest TAC values. This may just be a misunderstanding on my part but regardless some clarification would be helpful.

Also note that in addition to the TAC values for GRIP there is also TAC data for GISP2 in the Eemian. See (Yau, Bender, Robinson, & Brook, 2016).

Minor suggestions: Ln 20 – The integrity of the ice core record is not the issue as much

as our ability to interpret the records of CH4, N20 and TAC when they are affected by melt layers. Questioning the integrity of the ice core record makes me think something happened during analysis or storage of the ice core.

Ln 24 . . .only direct proxy for past surface elevation of the interior of an ice sheet.

Ln 25-26 It should also be mentioned that TAC responds on millennial time scales as noted by (Eicher et al., 2016). Their hypothesis is that these affects were due to rapid changes in accumulation. As this is unlikely to occur in the Eemian this can probably be ignored for the rest of the analysis but should still be mentioned here.

Ln 30 . . . TAC is the only direct method . . .

Ln 33 . . . NEEM-derived surface temperature anomaly relative to the last 1000 years . . .

Ln 115 -117 What assumptions are being made in using Henry's law to calculate air in melt water i.e. is the meltwater saturated with O2 and N2 when it is refrozen and are any bubbles incorporated into the meltwater? I think the assumption of using Henry's law works but the assumptions should be noted.

Ln 153 -Ln162 Melting and Refreezing and Tbl 1 Add accumulation to table 1. This will be useful when discussing refreeze relative to accumulation rates in the later section Ln 153-162. Alternatively plot the model accumulation rates.

Ln 204-206- This sentence is not clear to me. What part of the simulation would fit better?

Ln 257-258 – It is worth noting that if the lower TAC values in GRIP and NGRIP are related to surface melt rather than elevation, then CH4 over this period should be elevated with NGRIP being higher than GRIP and GISP2 due to the greater melt percent.

Readability:

There are sections of the paper that could use a revision for readability. The following

are a few examples:

Ln 6 can not -> cannot

Ln 20 . . . the presence of surface melt during ice formation can be a problem-> the presence of surface melt can be a problem

Ln 25: However, TAC was also found to have an insolation signal-> However, TAC is also affected by insolation at both Greenland and Antarctic sites (Eicher et al., 2016; Raynaud et al., 2007)

Ln 36 – unclear antecedent- Despite these concerns

Ln 62 moderate -> moderately

Ln 62? smaller Eemian ice sheet equivalent to ∼0.5 m of sea level rise.-> smaller Eemian ice sheet with the difference equivalent to ∼0.5 m of sea level rise

Ln 260 have an ensemble of climate to explore-> have an ensemble of climate models to explore

Final thoughts: I found this paper thought provoking. While I think there is work left to do, I look forward to seeing the next iteration.

Yau, A. M., Bender, M. L., Robinson, A., & Brook, E. J. (2016). Reconstructing the last interglacial at Summit, Greenland: Insights from GISP2. Proceedings of the National Academy of Sciences of the United States of America, 113(35), 9710-9715. doi:10.1073/pnas.1524766113

—————————————————————

---

## Referee Comment (RC2) · Anonymous Referee #2 · 12 Oct 2020

**Review** *"Greenland climate simulations show high Eemian surface melt"* by Plach et al.

The authors compare modeled and measured Eemian (130-115 ka) total air content (TAC) extracted from seven ice cores drilled in Greenland and the Canadian Arctic. TAC is a proxy commonly used to infer past changes in surface elevation since the density of air trapped in the ice declines with altitude. The authors show that low TAC values observed in Greenland Eemian ice are affected by high melt rates and subsequent refreezing that reduce TAC through the formation of ice layers (referred to as melt layers). Therefore, high Eemian melt rates could explain the low measured TAC in ice cores, a process that should be considered when estimating surface elevation changes in past warm periods.

The paper is well-written and provides important insights on the impact of Eemian high melt rates on measured TAC that should be accounted for to accurately estimate surface elevation changes in past warm periods. The paper would benefit from additional clarifications/details regarding the methods, model evaluation and study limitations. The reviewer deems that **minor revisions** are required before publication in *Climate of the Past*. The reviewer's comments are summarized hereunder.

**General comments**
1. The authors use the climate model MAR to dynamically downscale two Eemian time slices from the Earth System Model NorESM1-F (125 and 115 ka) as well as a pre-industrial control run. Modeled melt, refreezing and temperature are the core of the study as these are used to estimate modeled TAC that are compared with Eemian ice cores observations. The description of the MAR model is however not sufficient. The authors should mention which model version is used, and at what spatial resolution (i.e. 25 km in **L67** appears too late in the text). The authors should also briefly describe in Section 2 how surface melt (SEB-derived) and subsequent refreezing are calculated in MAR.
2. The authors prescribe a fixed contemporary Greenland ice sheet geometry in MAR to simulate the surface mass balance (SMB) components over the warmer than present Eemian period. This is acceptable given the lack of an accurate estimate of Eemian ice sheet geometry and the high computational costs of an offline coupling with an ice dynamics model (e.g. Le clec'h et al., 2019). However, the authors should discuss the limitations and uncertainties introduced by the use of a fixed modern ice sheet geometry. For instance, Van de Berg et al. (2011) and references therein suggest a 30-60% ice sheet volume reduction in the Eemian relative to present-day. Consequently, simulating melt and SMB on a more extensive, modern ice sheet may artificially cause high melt rates over larger ablation zones than expected if using a more accurate Eemian ice sheet geometry. Could the authors elaborate on this matter? Figure 1 could also show MAR melt rates averaged for the Eemian period 125 ka as a background.
3. The Eemian period is characterized by a climate significantly warmer than today, however in Fig. 2, annual mean near-surface temperature from the pre-industrial, 125 ka and 115 ka Eemian periods are almost systematically colder than or roughly equal to present-day observations. This is confusing especially since summer temperatures in the Eemian shown in Fig. 3 are considerably higher than present-day (3-4 K). Is this the result of a more pronounced seasonality of the Eemian climate, i.e. with colder winters and warmer summers, making the average annual temperature comparable to present-day but with markedly warmer summers? Could the authors further comment on this?

**Point comments**
**L6:** The reviewer suggests reformulating as: "Therefore, simulating high Eemian melt rates and associated melt layers is beneficial to improve the representation of past surface elevation."
**L23:** The authors could reformulate as: "However, refrozen melt has the potential to form impermeable ice layers (melt layers henceforth) that alter the diffusion of ice core signals."
**L33-35:** With respect to which period are these temperature anomalies estimated?
**L39:** The site GISP2 is not shown in Fig. 1 nor referred to elsewhere in the manuscript. The authors could remove "(used synonymous … proximity)."
**L40:** The authors could mention that Agassiz ice cap is situated in the northern Canadian Arctic.
**L42:** "evaluated" instead of "validated", same comment in **L50.** The authors should stress that present-day measurements are used as a reference for comparison with a warmer Eemian and colder

pre-industrial climate rather than for model "evaluation". Strictly speaking, present-day observations cannot be used to "validate" nor "evaluate" Eemian or pre-industrial climate.

**L47:** The reviewer suggests: "based on **two** Eemian time slice simulations … conditions and one pre-industrial (PI; **YYYY-YYYY**) control simulation." Later on in the text (**L52**) "four" Eemian experiments are mentioned while only two (125 and 115 ka) are described in the text. Please, mention the period spanned by the pre-industrial control run (e.g. 1850-1949?) as well as the 125 and 115 ka runs (i.e. number of thin lines in e.g. Fig. 2).

**L51:** Maybe "All climate simulations use a fixed, modern ice sheet geometry, in lack …" See also general comment #2, i.e. a too large ice sheet extent are likely to artificially increase surface melt.

**L54:** To clarify, the reviewer strongly suggests to replace "SEB-derived SMB" by "MAR SMB" across the manuscript.

**L56:** The authors could reformulate as: "Additionally, while providing the most complete representation of physical surface processes in the pool of investigated models, MAR shows lower Eemian melt rates (**XX**%) than intermediate complexity SMB models.".

**L62:** "Eemian ice sheet volume equivalent to ~0.5 m …"

**L71:** "SMB simulations are compared to present-day satellite …", see also comment in **L42**.

**L76:** The authors could reformulate as: "covers the whole MAR grid at 25 km from May to September for most years between 1979-2010".

**L93-100:** This paragraph describing the data sets presented in Figs. 6 and 7 should be moved to **P9** under Subsection *Total air content (TAC)*.

**L119:** In Eq. 6 "$C_{a,O2}$" instead of "$C_{a,N2}$".

**L124-126:** To the reviewer's knowledge, average pre-industrial temperatures should be colder than present-day observations. Could the authors elaborate on this?

**L128-129:** "The lower borehole … than near-surface temperature". The sentence is unclear, could the authors reformulate?

**L131-133:** This is confusing as temperature in the Eemian should be warmer and pre-indutrial temperature colder than present-day. For instance, how should readers interpret the fact that near-surface temperatures at NGRIP are systematically warmer in the pre-industrial period than in present-day? See also general comment #2

**L132:** "(Fig. 2; blue and orange)", there is no red data in Fig. 2.

**L134-135:** How come that the 3-4 K warming only appears in summer temperature, see also general comment #2.

**L138:** What do the authors mean by "precipitation-weighted temperatures"? How is this calculated? Why do annual precipitation-weighted temperatures show a warming similar to that of summer temperatures? What is the difference with the annual data shown in Fig. 2?

**L161:** The reviewer suggests: "~3,200 m elevation, refreezing surpasses 25% of the annual accumulation under 125 ka conditions. […] where refreezing percentages can reach 80-90%." It is much clearer to mention period averages (thick lines in Fig. 5) rather than single year values (thin lines).

**L167-168:** The authors should consider mentioning period averages as: "… 45-70 ml kg$^{-1}$ on average, whereas … between 75-100 ml kg$^{-1}$. At Dye-3 … is about 25 ml kg$^{-1}$ on average for the warm …"

**L173:** The authors should consider removing Dye-3 data in Fig. 7 as the ice core does not include Eemian ice.

**L196:** The reviewer suggests "the lowering and retreat of the Eemian ice sheet", see also general comment #2.

**L204-206:** This is unclear, could the authors reformulate?

**L214:** What do the authors mean by "100% melt"?

**L260-261:** Eemian melt derived from the regional climate model RACMO2 should be available from Van de Berg et al. (2011).

**L264-267:** Such analysis has been conducted in e.g. Fettweis et al. (2013) or Tedesco et al. (2020).

**L272:** The reviewer suggests: "The simulated air pressure … are used to estimate Eemian total air content (TAC). Simulated high melt rates could explain the low corresponding ice core TAC observations."

**Style**

**L3:** The reviewer suggests "affect" instead of "influence". Same in **L21** and **L44.**

**L5:** Do the authors mean "high surface melt" or "enhanced surface melt relative to present-day"?

**L9-10:** Replace "elevated levels of surface melt" by "high melt rates".
**L10:** "when interpreting measured Greenland TAC fluctuations as surface elevation changes."
**L19:** "favorable for high melt rates across the Greenland ice sheet."
**L20:** "alter" instead of "be a problem for".
**L26:** Replace "can be applied on" by "can be estimated for".
**L37:** "limited" instead of "small".
**L60:** "larger" instead of "bigger".
**L201:** "that the climate simulations might include a cold bias."
**L244:** "air content to estimate ice surface elevation changes".
**L259:** "obtain" instead of "accomplish".

**Figures**
**Fig. 1:** The authors could consider showing MAR Eemian melt as a background (125 ka).
**Figs. 2, 3, 5, 6 and A1-3**: Data should be shown in chronological order: PI (pre-industrial), 115 ka (late Eemian), and then 125 ka (early-Eemian).
**Fig. 4:** Replace "nan" by e.g. "NA" for "Not Available" and explain the acronym in the caption. NAN commonly means "Not A Number" while the authors certainly mean "unavailable data". How should readers interpret the fact that the number of melt days is higher in the present-day climate than in the warmer Eemian period at Agassiz site?
**Fig. 6 caption:** "almost completely overlaps with …".

**References**
Le clec'h et al. (2019): https://tc.copernicus.org/articles/13/373/2019/
Van de Berg et al. (2011): https://www.nature.com/articles/ngeo1245#Sec7
Fettweis et al. (2013): https://tc.copernicus.org/articles/7/241/2013/
Tedesco et al. (2020): https://tc.copernicus.org/articles/14/1209/2020/

---

## Author Comment (AC1) · 13 Nov 2020

**We would like to very much thank the anonymous referee #1 for reviewing our study and her/his constructive comments. Please find below the referee's comments in black font and** the authors' response in blue font.

The paper by Plach et al. addresses two topics of interest. First, it simulates the frequency of surface melt over Greenland during the Eemian from ~130ka to 115ka. The authors use the surface mass balance output from the Norwegian Earth System Model with time slices at 125 and 115ka in combination with the Modele Atmospherique to attain regional climates over Greenland. To test the viability of their approach, the models were validated by comparing them to preindustrial conditions. The authors conclude from the preindustrial comparison that their model is likely conservative in estimating the percentage of melt. They find that at 125ka the percentage of melt could exceed 25% of the annual accumulation at GRIP and up to 90% in less central locations such as Dye-3. These are impressive results and could be important for understanding seasonality effects on surface melt and on ice sheet mass balance.

Secondly Plach et al. addresses how surface melt can affect ice core records, particularly TAC. The authors point out that the interpretation of TAC as a unique proxy for elevation is complicated by the influence of surface melt during warm periods especially the Eemian where these layers cannot be visually identified due to thinning. Melt layers are generally bubble free with the amount of air in the melt layer being primarily dictated by Henry's law. The result is that melt layers have significantly lower TAC. Plach et al. use the results of their melt simulations to calculate theoretical Eemian TAC values which they call TACred and then compare these values to published data. The results of the model derived TAC is consistently lower than the measured data at 125ka but similar at 115ka and the preindustrial.

**General comments, suggestions and edits**

Both the extent of melt over Greenland, and how melt can affect TAC during warm periods are important topics and of value to the paleoclimate community. The approach of the authors to account for melt in TAC is novel and could help explain some of the anonymously low TAC values we see during the Eemian in some ice core records. I find that the subject matter fits well within the scope of Climate of the Past. However, the paper could use substantial revision, as noted below.

First, I wonder if it would be better to split this into two papers, one on the Greenland surface melt simulations and a second on the potential effects of melt on TAC records and their interpretation. This would enable a more detailed analysis of the surface melt simulations and the interpretation of them, as well as a more robust comparison between the derived or simulated TAC and the measured data. On both the melt simulation results and the derived TAC, I would like to see a more robust interpretation of the results. Assuming the authors choose to keep this as a single paper the following are my suggestions.
We agree that it would be interesting to perform further comparisons and more detailed analysis. However, since melt and TAC are strongly related, we prefer to keep the manuscript as a single paper. We address the reviewers concerns below.

**Major Suggestions**
Title: The title makes no mention of total air content but yet this is a major component of the

paper. I suggest changing the title to include total air content.

We agree, the total air content should also be represented in the title. Therefore, we will change the title to: "Greenland climate simulations show high Eemian surface melt explaining reduced total air content in ice cores"

Abstract: In contrast to the title the abstract is excessively focused on the TAC results. I suggest adding a few sentences about the melt simulations.

The current abstract illustrates the strong relation we see between melt and TAC, which is also a reason why we prefer to present both in one paper. However, we agree that the abstract is focused on TAC. We will add a few sentences on the melt simulations in the abstract giving a more consistent picture together with the new title.

Ln 101-120: Calculation of the model-derived total air content (TAC) The derived TAC does not account for insolation. As mentioned earlier in the paper (Eicher et al. 2016) show that TAC at high accumulation sites such as in Greenland have a similar imprint from insolation as in Antarctic records. The implication is that the comparison shown in figure 6 is between TAC records with the effects of insolation and derived values without the effect of insolation. This will bias the results. While this discrepancy is included in the Discussion section, it would be preferable here. Note that (Eicher et al. 2016) provides an insolation sensitivity (5.7 *10-9 mL kg-1 J-1) for the integrated insolation threshold over 390Wm-2. This should provide the framework to approximate how much insolation affects the analysis in this paper.

Eicher et al. (2016) analyzed the NGRIP ice core. To what degree their findings are representative for the entire Greenland Ice Sheet is an open question and goes beyond the scope of this paper. The dominant effect in our analysis is the melt effect not the insolation effect.

Ln 164-170: Results: Total air content (TAC) and Figure 7 In addition to comparing the derived results with data, it would be useful to compare the derived results with and without TACred included. This would help determine the magnitude of the effect on TAC. While the derived TAC may be lower than the measured TAC for a number of reasons including insolation effects and elevation uncertainty, the difference between derived TAC and TACred may still be fairly accurate and informative.

This will indeed be very informative. We will include a TAC-to-TACred comparison in the text and Fig. 6. Furthermore, we will consider also adding the comparison to Fig. 7.

Ln 95-96: and Figures 6 and 7 It is not clear to me why only the lowest 10% of TAC values (20% for GRIP) were used to determine the Eemian measured values for comparison to the derived values at the 125ka time slice. For the derived TAC the melt effect should be applicable to all TAC samples, not just the ones with the lowest TAC values. This may just be a misunderstanding on my part but regardless some clarification would be helpful.

The 125ka simulations represent the warmest part of the Eemian interglacial period in our analysis, we therefore expect the largest melt influence in this period. We choose the 10% (20%) lowest TAC values as a representation of the measurements most likely representing the strongest melt-influenced observations. We will add some clarification in the revised manuscript.

Also note that in addition to the TAC values for GRIP, there is also TAC data for GISP2 in the Eemian. See (Yau, Bender, Robinson, & Brook, 2016).

Thank you for this reference. We will consider adding the GISP2 TAC data to the revised manuscript. Alternatively, we will discuss its characteristics in the analysis.

**Minor suggestions**

Ln 20 – The integrity of the ice core record is not the issue as much as our ability to interpret the records of CH4, N20 and TAC when they are affected by melt layers. Questioning the integrity of the ice core record makes me think something happened during analysis or storage of the ice core.
Thank you for this note. We will reformulate in the revised manuscript.

Ln 24 . . .only direct proxy for past surface elevation of the interior of an ice sheet.

Ln 25-26 It should also be mentioned that TAC responds on millennial time scales as noted by (Eicher et al., 2016). Their hypothesis is that these effects were due to rapid changes in accumulation. As this is unlikely to occur in the Eemian this can probably be ignored for the rest of the analysis but should still be mentioned here.

Ln 30 . . . TAC is the only direct method . . .

Ln 33 . . . NEEM-derived surface temperature anomaly relative to the last 1000 years…
Thank you, we will address the points above in the revised manuscript.

Ln 115 -117 What assumptions are being made in using Henry's law to calculate air in melt water i.e. is the meltwater saturated with O2 and N2 when it is refrozen and are any bubbles incorporated into the meltwater? I think the assumption of using Henry's law works but the assumptions should be noted.
To calculate TACrefrozen, it is assumed that the meltwater is in equilibrium with air at a temperature of 273K and at the local atmospheric pressure (Eq. 5 and 6 in the discussion paper). No air is occluded in the form of bubbles in the freezing process. We will revise the TAC method section and clarify the assumptions.

Ln 153 -Ln162 Melting and Refreezing and Tbl 1 Add accumulation to table 1. This will be useful when discussing refreeze relative to accumulation rates in the later section

Ln 153-162. Alternatively plot the model accumulation rates.
Thank you for this suggestion. We will add the accumulation to Table 1.

Ln 204-206- This sentence is not clear to me. What part of the simulation would fit better?
We are referring to the large uncertainty of the NEEM temperature reconstructions (warming of 8°C +/-4°C) which is largely related to the uncertain elevation change. The lower end of the NEEM reconstructions would fit better with the simulated warming of ~3-4°C found in our study. We will clarify this sentence in the revised manuscript.

Ln 257-258 – It is worth noting that if the lower TAC values in GRIP and NGRIP are related to surface melt rather than elevation, then CH4 over this period should be elevated with NGRIP being higher than GRIP and GISP2 due to the greater melt percent.
While the referee is correct in the above statement, we are not aware that data to confirm

this is available. However, we will consider adding a note on this in the revised manuscript.

**Readability**

There are sections of the paper that could use a revision for readability. The following are a few examples:

Ln 6 can not -> cannot

Ln 20 . . . the presence of surface melt during ice formation can be a problem -> the presence of surface melt can be a problem

Ln 25: However, TAC was also found to have an insolation signal-> However, TAC is also affected by insolation at both Greenland and Antarctic sites (Eicher et al., 2016; Raynaud et al., 2007)

Ln 36 – unclear antecedent- Despite these concerns

Ln 62 moderate -> moderately

Ln 62? smaller Eemian ice sheet equivalent to ~0.5 m of sea level rise.-> smaller Eemian ice sheet with the difference equivalent to ~0.5 m of sea level rise

Ln 260 have an ensemble of climate to explore-> have an ensemble of climate models to explore

We will address the named readability issues and give the manuscript another round of proofreading to improve the readability.

**Final thoughts**

I found this paper thought provoking. While I think there is work left to do, I look forward to seeing the next iteration.

Yau, A. M., Bender, M. L., Robinson, A., & Brook, E. J. (2016). Reconstructing the last interglacial at Summit, Greenland: Insights from GISP2. Proceedings of the National Academy of Sciences of the United States of America, 113(35), 9710-9715. doi:10.1073/pnas.1524766113

Thank you very much for your overall positive feedback. Your comments and suggestions will help to significantly improve our manuscript.

---

## Author Comment (AC2) · 13 Nov 2020

**We would like to very much thank the anonymous referee #2 for reviewing our study and her/his constructive comments. Please find below the referee's comments in black font and** the authors' response in blue font.

**Review** "Greenland climate simulations show high Eemian surface melt" by Plach et al.

The authors compare modeled and measured Eemian (130-115 ka) total air content (TAC) extracted from seven ice cores drilled in Greenland and the Canadian Arctic. TAC is a proxy commonly used to infer past changes in surface elevation since the density of air trapped in the ice declines with altitude. The authors show that low TAC values observed in Greenland Eemian ice are affected by high melt rates and subsequent refreezing that reduce TAC through the formation of ice layers (referred to as melt layers). Therefore, high Eemian melt rates could explain the low measured TAC in ice cores, a process that should be considered when estimating surface elevation changes in past warm periods.

The paper is well-written and provides important insights on the impact of Eemian high melt rates on measured TAC that should be accounted for to accurately estimate surface elevation changes in past warm periods. The paper would benefit from additional clarifications/details regarding the methods, model evaluation and study limitations. The reviewer deems that **minor revisions** are required before publication in Climate of the Past. The reviewer's comments are summarized hereunder.

**General comments**

1. The authors use the climate model MAR to dynamically downscale two Eemian time slices from the Earth System Model NorESM1-F (125 and 115 ka) as well as a pre-industrial control run. Modeled melt, refreezing and temperature are the core of the study as these are used to estimate modeled TAC that are compared with Eemian ice cores observations. The description of the MAR model is however not sufficient. The authors should mention which model version is used, and at what spatial resolution (i.e. 25 km in **L67** appears too late in the text). The authors should also briefly describe in Section 2 how surface melt (SEB-derived) and subsequent refreezing are calculated in MAR.
   Thank you. We will extend the description of the MAR model and address your suggestions in the revised manuscript.

2. The authors prescribe a fixed contemporary Greenland ice sheet geometry in MAR to simulate the surface mass balance (SMB) components over the warmer than present Eemian period. This is acceptable given the lack of an accurate estimate of Eemian ice sheet geometry and the high computational costs of an offline coupling with an ice dynamics model (e.g. Le clec'h et al., 2019). However, the authors should discuss the limitations and uncertainties introduced by the use of a fixed modern ice sheet geometry. For instance, Van de Berg et al. (2011) and references therein suggest a 30-60% ice sheet volume reduction in the Eemian relative to present-day. Consequently, simulating melt and SMB on a more extensive, modern ice sheet may artificially cause high melt rates over larger ablation zones than expected if using a more accurate Eemian ice sheet geometry. Could the authors elaborate on this matter? Figure 1 could also show MAR melt rates averaged for the Eemian period

125 ka as a background.

We agree that there is a large uncertainty in the Eemian ice sheet geometry and that a model ice sheet geometry, which is too large during the melt simulations, could cause artificially high melt rates. However, such artificially high melt rates will influence regions on the margins much stronger than sites in the ice sheet center which are the main focus of our study. Ideally, a more systematic evaluation of Eemian melt at the ice cores sites should also investigate different possible ice sheet geometries. However, this is, as you mention, difficult due to high computational costs and not within the scope of our initial investigation in this study. We will include a discussion of these points in the revised manuscript with a focus on the uncertainty in using a given ice sheet geometry.

Furthermore, we will consider adding the 125k melt as a background of Fig. 1.

3. The Eemian period is characterized by a climate significantly warmer than today, however in Fig. 2, annual mean near-surface temperature from the pre-industrial, 125 ka and 115 ka Eemian periods are almost systematically colder than or roughly equal to present-day observations. This is confusing especially since summer temperatures in the Eemian shown in Fig. 3 are considerably higher than present-day (3-4 K). Is this the result of a more pronounced seasonality of the Eemian climate, i.e. with colder winters and warmer summers, making the average annual temperature comparable to present-day but with markedly warmer summers? Could the authors further comment on this?

You are absolutely right, the Eemian interglacial period was characterized by a more pronounced seasonality due to the difference in the Earth's orbital parameters (larger obliquity and eccentricity; Yin and Berger, 2010): giving a positive summer insolation anomaly and warmer-than-present summers in the Northern Hemisphere, as also recorded in Greenland ice cores and elsewhere in the Arctic (CAPE Last Interglacial Project Members, 2006). We will clarify this point in the revised manuscript.

Yin, Q. Z. and Berger, A.: Insolation and CO2 contribution to the interglacial climate before and after the Mid-Brunhes Event, Nature Geoscience, 3, 243–246, https://doi.org/10.1038/ngeo771, 2010.

CAPE Last Interglacial Project Members: Last Interglacial Arctic warmth confirms polar amplification of climate change, Quaternary Science Reviews, 25, 1383–1400, https://doi.org/10.1016/j.quascirev.2006.01.033, 2006.

**Point comments**

**L6:** The reviewer suggests reformulating as: "Therefore, simulating high Eemian melt rates and associated melt layers is beneficial to improve the representation of past surface elevation."

**L23:** The authors could reformulate as: "However, refrozen melt has the potential to form impermeable ice layers (melt layers henceforth) that alter the diffusion of ice core signals."

We will modify the two sentences above accordingly.

**L33-35:** With respect to which period are these temperature anomalies estimated?

The cited temperature anomalies are relative to the mean of the past millennium. We will add this in the text.

**L39:** The site GISP2 is not shown in Fig. 1 nor referred to elsewhere in the manuscript. The authors could remove "(used synonymous ... proximity)."
We will revisit the discussion of GISP2 as referee #1 pointed out existing Eemian TAC data for GISP2.

**L40:** The authors could mention that Agassiz ice cap is situated in the northern Canadian Arctic.
We will add this information.

**L42:** "evaluated" instead of "validated", same comment in **L50**. The authors should stress that present-day measurements are used as a reference for comparison with a warmer Eemian and colder pre-industrial climate rather than for model "evaluation". Strictly speaking, present-day observations cannot be used to "validate" nor "evaluate" Eemian or pre-industrial climate.
We will reformulate this section. We think that the word "validated" is used correctly in L50, as the MAR model has been shown to be able to represent the present-day climate well over Greenland in several studies. We will clarify that we refer to a validation under present-day climate conditions by modifying L50 to "which was extensively validated over Greenland under present-day climate conditions".

**L47**: The reviewer suggests: "based on **two** Eemian time slice simulations ... conditions and one preindustrial (PI; **YYYY-YYYY**) control simulation." Later on in the text (**L52**) "four" Eemian experiments are mentioned while only two (125 and 115 ka) are described in the text. Please, mention the period spanned by the pre-industrial control run (e.g. 1850-1949?) as well as the 125 and 115 ka runs (i.e. number of thin lines in e.g. Fig. 2).
The global NorESM-F runs are started with a 1000 year equilibrium pre-industrial run (pre-industrial refers to constant 1850 forcing; GHG and orbital parameters). After this the pre-industrial run was continued for another 1000 years (with constant 1850 forcing). Additionally, after the first 1000 year equilibrium run the Eemian runs are branched off and run for 1000 years with constant Eemian forcing (115 and 125ka, respectively; changed GHG and orbital parameters). The downscaling with MAR is done with the last 30 years of the NorESM simulations, while the first 4 years are used as a spin-up for MAR and not used in the analysis. Therefore, the analysis of the pre-industrial (constant 1850 forcing) and Eemian melt simulations (constant 115 and 125ka forcing) are based on 26 years of MAR simulations.
We will include this information in the revised manuscript.

**L51**: Maybe "All climate simulations use a fixed, modern ice sheet geometry, in lack ..." See also general comment #2, i.e. a too large ice sheet extent are likely to artificially increase surface melt.

**L54**: To clarify, the reviewer strongly suggests to replace "SEB-derived SMB" by "MAR SMB" across the manuscript.
Yes, we will do that.

**L56**: The authors could reformulate as: "Additionally, while providing the most complete representation of physical surface processes in the pool of investigated models, MAR shows lower Eemian melt rates (**XX**%) than intermediate complexity SMB models.".

**L62**: "Eemian ice sheet volume equivalent to ~0.5 m ..."

**L71**: "SMB simulations are compared to present-day satellite ...", see also comment in **L42**.

**L76**: The authors could reformulate as: "covers the whole MAR grid at 25 km from May to September for most years between 1979-2010".

**L93-100**: This paragraph describing the data sets presented in Figs. 6 and 7 should be moved to **P9** under Subsection *Total air content (TAC).*

**L119**: In Eq. 6 "$C_{a,O2}$" instead of "$C_{a,N2}$".
Thank you, we will change the manuscript according to the suggestions above.

**L124-126**: To the reviewer's knowledge, average pre-industrial temperatures should be colder than present-day observations. Could the authors elaborate on this?
The observations from weather stations used for the comparisons of observed and simulated annual mean temperature (Fig. 2, long black bold line) cover the period 1890 to 2014. You are right that we would expect the present-day annual mean temperatures to be higher than pre-industrial temperatures (at least for the last few decades). However, the long averaging period from 1890 to 2014 should reduce the influence of recent global warming. Furthermore, the warmer-than-present simulated pre-industrial temperatures indicate that the climate simulations are conservative in terms of temperature, or at least not particularly warm, and therefore should not result in extreme melt.
In the revised manuscript we will clarify the difference between the temperatures presented in Figs. 2. and 3 (averaged over the period 1890-2014) and the pre-industrial and present day temperatures.

**L128-129**: "The lower borehole ... than near-surface temperature". The sentence is unclear, could the authors reformulate?
We will reformulate this sentence.

**L131-133**: This is confusing as temperature in the Eemian should be warmer and pre-industrial temperature colder than present-day. For instance, how should readers interpret the fact that near-surface temperatures at NGRIP are systematically warmer in the pre-industrial period than in present-day? See also general comment #2
You are right, this is confusing. We will clarify this section in the revised manuscript. We don't expect the annual mean temperatures for pre-industrial and Eemian to be very different (since the total amount of solar insolation didn't differ much). However, since the Eemian seasonality was much more pronounced the Eemian summer temperatures should be higher than the pre-industrial ones as seen in Fig. 3 (shown the simulated JJA temperatures).

**L132**: "(Fig. 2; blue and orange)", there is no red data in Fig. 2.
The red color is a remainder of a previous manuscript version. We will remove this reference.

**L134-135**: How come that the 3-4 K warming only appears in summer temperature, see also general comment #2.

Due to the higher Eemian seasonality. Also see our response to general comment #3.

**L138**: What do the authors mean by "precipitation-weighted temperatures"? How is this calculated? Why do annual precipitation-weighted temperatures show a warming similar to that of summer temperatures? What is the difference with the annual data shown in Fig. 2?

For the calculation of the precipitation-weighted temperature, daily temperatures are multiplied by the precipitation (snowfall+rainfall) at the individual days, summed up over the year and then divided by the sum of the annual precipitation. Precipitation is used as a weight, instead of time in the usual averages where each time step is equally represented in the average. In the precipitation-weighted temperature, days with a lot of precipitation are weighted stronger than days with low precipitation, and days with no precipitation are not represented. The precipitation-weighted temperature is sometimes used in the interpretation of ice core temperature reconstructions, since ice cores can only record temperatures if there is some kind of precipitation deposited. Since most precipitation in Greenland falls in summer, the precipitation-weighted temperature is more similar to summer temperatures than it is to annual mean temperature. We will clarify this in the revised manuscript.

**L161**: The reviewer suggests: "~3,200 m elevation, refreezing surpasses 25% of the annual accumulation under 125 ka conditions. [...] where refreezing percentages can reach 80-90%." It is much clearer to mention period averages (thick lines in Fig. 5) rather than single year values (thin lines).

**L167-168**: The authors should consider mentioning period averages as: "... 45-70 ml kg-1 on average, whereas ... between 75-100 ml kg-1. At Dye-3 ... is about 25 ml kg-1 on average for the warm ..."

We will adapt the two sections above accordingly in the revised manuscript.

**L173**: The authors should consider removing Dye-3 data in Fig. 7 as the ice core does not include Eemian ice.

Although, there are no Eemian TAC measurements for Dye-3. The ice at the bottom of Dye-3 has been found to be much older than the Eemian interglacial period (Willerslev et al., 2007). Furthermore, Dye-3 illustrates how our melt/TAC calculations play out at a more marginal site. Therefore, we prefer to include the TAC observations at this site.

Willerslev E, Cappellini E, Boomsma W, et al. (2007): Ancient biomolecules from deep ice cores reveal a forested southern Greenland. Science. 317(5834):111-114. doi:10.1126/science.1141758

**L196**: The reviewer suggests "the lowering and retreat of the Eemian ice sheet", see also general comment #2.

**L204-206**: This is unclear, could the authors reformulate?

We will reformulate the two sections above accordingly in the revised manuscript.

**L214**: What do the authors mean by "100% melt"?

This should actually refer to a refreezing percentage of 100%. We will revise this sentence.

**L260-261**: Eemian melt derived from the regional climate model RACMO2 should be available from Van de Berg et al. (2011).

This is a very good point, that other melt data from other Eemian simulations might be available. However, we see our analysis as an initial study investigating the relationship between melt and TAC during the Eemian interglacial period. A more systematic analysis comparing the output of different climate models is a possible work for the future.

**L264-267**: Such analysis has been conducted in e.g. Fettweis et al. (2013) or Tedesco et al. (2020).

Thank you, we will mention this in the revised discussion.

**L272**: The reviewer suggests: "The simulated air pressure ... are used to estimate Eemian total air content (TAC). Simulated high melt rates could explain the low corresponding ice core TAC observations."

Thank you, we will change this sentence accordingly.

**Style**

**L3**: The reviewer suggests "affect" instead of "influence". Same in **L21** and **L44**.
**L5**: Do the authors mean "high surface melt" or "enhanced surface melt relative to present-day"?
**L9-10**: Replace "elevated levels of surface melt" by "high melt rates".
**L10**: "when interpreting measured Greenland TAC fluctuations as surface elevation changes."
**L19**: "favorable for high melt rates across the Greenland ice sheet."
**L20**: "alter" instead of "be a problem for".
**L26**: Replace "can be applied on" by "can be estimated for".
**L37**: "limited" instead of "small".
**L60**: "larger" instead of "bigger".
**L201**: "that the climate simulations might include a cold bias."
**L244**: "air content to estimate ice surface elevation changes".
**L259**: "obtain" instead of "accomplish".

Thank you, we will revise the corresponding lines.

**Figures**

**Fig. 1**: The authors could consider showing MAR Eemian melt as a background (125 ka).
**Figs. 2, 3, 5, 6 and A1-3**: Data should be shown in chronological order: PI (pre-industrial), 115 ka (late Eemian), and then 125 ka (early-Eemian).
**Fig. 4**: Replace "nan" by e.g. "NA" for "Not Available" and explain the acronym in the caption. NAN commonly means "Not A Number" while the authors certainly mean "unavailable data". How should readers interpret the fact that the number of melt days is higher in the present-day climate than in the warmer Eemian period at Agassiz site?
**Fig. 6 caption**: "almost completely overlaps with ...".

We will consider your suggestions for the adaptation of the figures in the revised manuscript. Furthermore, we will adapt the discussion of the observed vs. Eemian melt at the Agassiz

site.

**References**

Le clec'h et al. (2019): https://tc.copernicus.org/articles/13/373/2019/
Van de Berg et al. (2011): https://www.nature.com/articles/ngeo1245#Sec7
Fettweis et al. (2013): https://tc.copernicus.org/articles/7/241/2013/
Tedesco et al. (2020): https://tc.copernicus.org/articles/14/1209/2020/

Thank you very much for your overall positive feedback. Your comments and suggestions will help to significantly improve our manuscript.